# Contextual Multi-Armed Bandits with Minimum Aggregated Revenue Constraints

**Ahmed Ben Yahmed**[1,2,3*]**, Hafedh El Ferchichi**[2,3]**, Marc Abeille**[1,2,3] **& Vianney Perchet**[1,2,3]
[1] Criteo AI Lab, Paris, France
[2] CREST, ENSAE, Institut Polytechnique de Paris, France
[3] FairPlay joint team, Inria, France

## Abstract

We examine a multi-armed bandit problem with contextual information, where the objective is to ensure that each arm receives a minimum aggregated reward across contexts while simultaneously maximizing the total cumulative reward. This framework captures a broad class of real-world applications where fair revenue allocation is critical and contextual variation is inherent. The cross-context aggregation of minimum reward constraints, while enabling better performance and easier feasibility, introduces significant technical challenges—particularly the absence of closed-form optimal allocations typically available in standard MAB settings. We design and analyze algorithms that either optimistically prioritize performance or pessimistically enforce constraint satisfaction. For each algorithm, we derive problem-dependent upper bounds on both regret and constraint violations. Furthermore, we establish a lower bound demonstrating that the dependence on the time horizon in our results is optimal in general and revealing fundamental limitations of the free exploration principle leveraged in prior work.

## 1 Introduction

The Multi-Armed Bandit (MAB) problem provides a foundational model for sequential decision-making under uncertainty (Thompson, 1933; Lattimore and Szepesvári, 2020; Auer et al., 2002; Bubeck and Cesa-Bianchi, 2012). At each step of a $T$ period run, an agent selects one of $K$ actions (arms), each yielding stochastic rewards, with the goal of maximizing cumulative reward. A central challenge is to balance *exploration*—gathering information about unknown rewards—and *exploitation*—leveraging current knowledge to optimize performance. Many variants and extensions of the synthetic bandit framework have been proposed to address specific challenges arising in real-world applications. In particular, for clinical trials, stringent safety constraints require the selection of treatment–dosage combinations that balance efficacy with the mitigation of adverse effects (Chen et al., 2022b; Pacchiano et al., 2021; Amani et al., 2019). Similarly, budget-constrained scenarios give rise to *knapsack bandits*, where the objective is to maximize cumulative rewards while adhering to a fixed resource allocation (Badanidiyuru et al., 2018; Chzhen et al., 2023). Additionally, fairness considerations may impose further constraints, such as ensuring equitable exposure across arms (Wang et al., 2021; Li et al., 2020) or guaranteeing minimum revenue thresholds for each arm (Baudry et al., 2024). Those settings require to extending the MAB framework to accommodate with reward maximization under various constraints.

We investigate a contextual MAB problem subject to per-arm minimum revenue guarantees. The learner's objective is to maximize the cumulative reward over time while ensuring that, on average, each arm $k$ achieves a reward of at least $\lambda_k$, a predefined minimum aggregated reward over all contexts. The learner must balance the trade-off between selecting the best arm in a given context and favoring a suboptimal arm to ensure it meets its minimum revenue requirement. As illustrated in Figure 1, depending on the problem parameters, different regimes can arise: (i) *infeasibility*, where the constraints cannot be satisfied; (ii) *feasibility with high cost*, where satisfying the constraints requires playing significantly suboptimal arms; and (iii) *feasibility with moderate cost*, where the performance gap is small and balancing reward and constraint satisfaction is relatively easy. This rich setting is motivated by several real-world applications. For instance, consider a movie recommendation platform that collaborates with multiple content providers. Each provider offers a catalog of movies spanning various categories, such as action, romance, and comedy. Users interact with the platform

---

*Correspondence to: a.benyahmed@criteo.com.

by selecting a category, and the system recommends a movie accordingly. While the platform aims to match users with the most relevant content (i.e., to maximize the reward), it must also ensure that each provider receives a minimum level of user engagement or revenue. This guarantee is essential to maintain providers' incentives for participating in the platform.

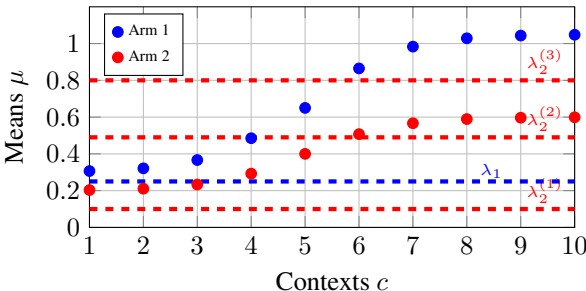

Figure 1: Illustration of a MAB problem with minimum aggregated reward constraints with two arms and multiple contexts. Arm 1 is optimal in all contexts. We fix the threshold $\lambda_1$ for arm 1 and we consider how different thresholds $\lambda_2^{(\cdot)}$ for arm 2 change the problem. If $\lambda_2 = \lambda_2^{(3)}$, the problem becomes infeasible. If $\lambda_2 = \lambda_2^{(2)}$, then arm 2 must be played frequently in contexts $c \geq 6$, which substantially reduces overall performance due to the large performance gap. However, if $\lambda_2 = \lambda_2^{(1)}$, it is sufficient to play arm 2 in contexts $c \leq 5$, where the performance gap is small, thus preserving overall reward.

## 1.1 RELATED WORK

Motivated by the demands of real-world applications, several extensions of the MAB framework have been proposed (hun Kim et al., 2024; Ben Yahmed et al., 2024a;b). The problem studied in this work belongs to the broader class of constrained bandit problems, which have been investigated under various motivations such as safety (Chen et al., 2022b), fairness (Wang et al., 2021), return-on-spend guarantees (Feng et al., 2023), conservative behavior (Wu et al., 2016; Deb et al., 2025), and knapsack constraints (Badanidiyuru et al., 2018; Bernasconi et al., 2024).

Constrained bandit problems has been studied under two distinct perspectives: a first stream of research focuses on hard constraints where violations are strictly prohibited, for instance in the linear constrained bandits setting (see (Pacchiano et al., 2021; Amani et al., 2019)). These approaches require prior knowledge of a feasible (safe) policy and employ carefully constructed pessimistic confidence sets to maintain zero constraint violation while achieving $\mathcal{O}(\sqrt{T})$ regret. An alternative approach relaxes the initial safe action requirement, allowing round-wise constraint violations while studying the fundamental trade-off between performance and constraint satisfaction, the two central metrics in constrained bandit problems. Notably, (Chen et al., 2022b) developed two algorithms for the non-contextual MAB setting: the first achieves $\mathcal{O}(\sqrt{T})$ regret with logarithmic constraint violation, while the second exhibits the inverse behavior. (Gangrade et al., 2024) introduced an algorithm for safe linear bandits to learn the optimal action defined by $\max_{x \in \mathbb{R}^d} x^\top \theta^\star$ s.t. $Ax \leq b$, achieving poly-logarithmic regret with $\mathcal{O}(\sqrt{T})$ constraint violation.

Bandits with knapsacks (BwK) have been extensively studied (Tran-Thanh et al., 2012; Badanidiyuru et al., 2014; Sivakumar et al., 2022; Kumar and Kleinberg, 2022; Han et al., 2023; Slivkins et al., 2024; Guo and Liu, 2025). Notably, Agrawal et al. (2016) extend BwK to contextual settings using the optimization framework of Agarwal et al. (2014), incorporating constraints via a primal-dual approach to achieve $\mathcal{O}(\sqrt{T})$ regret. Chen et al. (2024) consider a related contextual decision-making framework with knapsacks, but under a different feedback structure. In their model, reward and cost functions are known and depend on a random request, an external random variable, and the chosen action. Building elegantly on network revenue management techniques (Chen et al., 2022a; Balseiro et al., 2024), they employ a re-solving method conceptually similar to ours, providing guarantees for both full and partial information regimes, including beyond-worst-case analysis. A fundamental distinction between our setting and BwK lies in the nature of constraints: in BwK, costs accumulate until a fixed budget is exhausted, ensuring hard constraint satisfaction, whereas our constraints are stochastic and enforced in expectation, permitting explicit trade-offs between regret and constraint violation.

A special case of our setting is the context-free MAB with per-arm revenue guarantees studied in Baudry et al. (2024). Here, the optimal strategy pulls all arms linearly, proportional to their guarantee-to-performance ratio. The authors propose learning strategies using optimistic/pessimistic estimators: optimism improves rewards but increases constraint violation, while pessimism has the opposite effect. They demonstrate a trade-off between constant regret with $\mathcal{O}(\sqrt{T})$ violation or vice versa, improving on Chen et al. (2022b). This constant regret is possible due to free exploration from the optimal allocation's structure.

A generalization of our setting is the stochastic contextual bandit problem under general constraints, considered by (Slivkins et al., 2023). Assuming strict feasibility characterized by a known Slater constant $\gamma^\star$, they introduce a primal-dual optimization algorithm and establish $\mathcal{O}(\sqrt{T}/\gamma^\star)$ upper bounds on both regret and constraint violation. (Guo and Liu, 2024) extend this setting by removing the Slater condition, proving $\mathcal{O}(T^{3/4})$ bounds for both regret and constraint violation. Furthermore, when Slater's condition does hold, their method achieves improved bounds of $\mathcal{O}(\sqrt{T}/\gamma^{\star 2})$, notably without requiring prior knowledge of $\gamma^\star$. The recent work of (Guo et al., 2025) considers exactly the same setting and assumptions as (Guo and Liu, 2024), but proposes an algorithm that integrates a primal-dual approach with optimistic estimation, yielding $\mathcal{O}(\sqrt{T})$ bounds on both metrics.

## 1.2 CHALLENGES AND CONTRIBUTIONS

How to leverage contextual information while preserving revenue guarantee for each arm is a challenging (and open) question (Baudry et al., 2024). The reason is that most contextual learning problems can be reduced to a family of "local" independent problems. For instance, a contextual bandit problem reduces to many standard multi-armed bandits (Perchet and Rigollet, 2013), where the optimal decision can be computed based solely on the reward functions, context by context. With revenue constraints, this would be possible if the latter were defined context-wise – i.e., by specifying $\lambda_{k,c}$ for all pairs $(k, c)$, and would result in straightforward extension (by considering $|\mathcal{C}|$ parallel instances). Unfortunately, with aggregated constraints, this local reduction is impossible: it is not enough to learn that an arm is sub-optimal for a given context; what really matters is *how much* sub-optimal it is compared to others, so that a global planning can be computed. Even the planning problem becomes more complex with global constraints; we shall show that it can be reduced to solving a *linear program*, which preserves computational efficiency but sacrifices the closed-form solution that played a central role in the theoretical analysis of (Baudry et al., 2024).

While the works of (Slivkins et al., 2023; Guo and Liu, 2024; Guo et al., 2025) address contextual MAB problems that subsume our setting, we claim that their primal-dual approach—which guarantees $\mathcal{O}(\sqrt{T})$ bounds for both performance and constraint regret— is suboptimal in our case, as it overlooks the finer structure inherent to our problem formulation.

Consequently, our work is the first to bridge the gap between the non-contextual MAB with minimum revenue constraints studied in (Baudry et al., 2024), and the contextual MAB frameworks with general stochastic constraints explored in (Slivkins et al., 2023; Guo and Liu, 2024; Guo et al., 2025). We achieve this by introducing a novel approach that circumvents the absence of a closed-form solution to the planning problem, without relying on the primal-dual methodology. Our main contributions are the following:

1. We introduce two novel algorithms, **OLP** and **OPLP**, that seamlessly integrate linear programming with optimistic and pessimistic estimation techniques. These algorithms effectively navigate the trade-off between performance and constraint satisfaction, each capturing distinct points along the Pareto frontier of these competing objectives. They provably achieve poly-logarithmic regret with $\mathcal{O}(\sqrt{T})$ constraint violation, and vice versa.

2. The analytical techniques employed in this work are non-standard and may be of independent interest to the MAB literature. In particular, we derive poly-logarithmic bounds by introducing a novel and more refined notion of the sub-optimality gap, which leverages the structure of the underlying linear program and quantifies the complexity of learning the optimal policy. The proposed methodology extends naturally to a wide range of MAB problems that require enforcing global linear constraints.

3. We establish a lower bound that confirms the (near) optimality of our algorithms and we highlight a more intriguing interpretation of the exploration-exploitation trade-off in MAB with revenue guarantees. While non-contextual setting enjoys a *free-exploration* property

inherited from the constrained structure, this no longer holds in the contextual setting, where the exploration-exploitation trade-off is reinstated.

## 2 PROBLEM STATEMENT

**Notations** Let $a$ denote a generic quantity of interest. We use the notation $a_{k,c} \in \mathbb{R}$ to represent the value associated with arm $k$ in context $c$. The vector $\boldsymbol{a}_c = (a_{k,c})_{k \in \{1,\dots,K\}}$ in $\mathbb{R}^K$ collects these values across arms for a fixed context $c$, and the matrix $\underline{\boldsymbol{a}} = (a_{k,c})_{k \in \{1,\dots,K\}, c \in \mathcal{C}}$ in $\mathbb{R}^{K \times |\mathcal{C}|}$ gathers all arm-context values. We denote by $a_{k,c}(t)$, $\boldsymbol{a}_c(t)$, and $\underline{\boldsymbol{a}}(t)$ the time-dependent versions of these quantities at round $t$. We denote by $\boldsymbol{e}_{kk}$ the matrix in $\mathbb{R}^{K \times K}$ with a 1 in the $(k,k)$-th position and zeros elsewhere, and by $\boldsymbol{e}_k$ the $k$-th canonical basis vector in $\mathbb{R}^K$. The set of arms is $\mathcal{K} = \{1,\dots,K\}$, and the set of arm-context pairs is $\mathcal{J} = \{(k,c) \mid k \in \mathcal{K}, c \in \mathcal{C}\}$, with total cardinality $\kappa = K|\mathcal{C}|$. Finally, $\pi_K$ denotes the $K$-dimensional probability simplex, $\pi_K^{|C|}$ denotes the set of $K \times |C|$ matrices whose columns each belong to the simplex $\pi_K$, $(x)_+ = \max(x, 0)$ represents the positive part of $x$, and $|\mathcal{S}|$ stands for the cardinality of a set $\mathcal{S}$.

### 2.1 SETTING

We study a multi-armed bandit problem involving $K$ stochastic arms and a number of contextual scenarios. Let $\mathcal{C}$ denote the set of all possible contexts where each $c \in \mathcal{C}$ occurs with probability $p_c$. The expected reward of arm $k$ in a given context $c$ is represented by $\mu_{k,c}$.

At each time step $t$, the learner observes a context $c_t$, selects an arm $k_t \in \mathcal{K}$ and receives a reward $r_t$, which is independently drawn from the distribution $\mathcal{D}_{k_t, c_t}$. At each time $t$, the choice of the arm is based on history of past interactions $\mathcal{H}_{t-1} = (c_1, k_1, r_1, \dots, c_{t-1}, k_{t-1}, r_{t-1})$ and current context $c_t$. The interaction structure of this Multi-Armed Bandit with Aggregated Revenue Constraints (**MAB-ARC**) is summarized on the right.

---

**MAB-ARC:** MAB with Aggregated Revenue Constraints

---

**Inputs:** $\{\lambda_k\}_{k \in \mathcal{K}}$
**for** $t = 1$ **to** $T$ **do**
  Observe context $c_t$
  Choose arm $k_t$
  Receive reward $r_t \sim \mu_{k_t, c_t}$
  Update $\mathcal{H}_t = \mathcal{H}_{t-1} \cup \{c_t, k_t, r_t\}$

---

The learner aims to maximize the expected cumulative revenue over $T$ time steps while ensuring that the expected aggregated revenue from each arm $k$ over all contexts is larger than a predefined threshold $\lambda_k$.

**Planning Problem.** Formally, given known thresholds $\lambda_k$ and mean reward $\underline{\boldsymbol{\mu}}$, the optimization problem is defined as:

$$\mathbf{OBJ}: \max_{\{k_t\}_{t:1,\dots,T}} \mathbb{E}\left[\sum_{t=1}^T r_t\right] \quad \text{subject to: } \forall k \in \mathcal{K}, \ \mathbb{E}\left[\sum_{t=1}^T r_t \mathbb{1}_{[k_t = k]}\right] \geq \lambda_k T.$$

Optimal solutions to this constrained optimization problem consist in policies, i.e. allocation rules that map the current context to the probability of sampling an arm. We summarize allocation rules by $\underline{\boldsymbol{w}}$, where $w_{k,c} = \mathbb{P}(k|c)$. Interestingly, there is in general no unique optimal solution to **OBJ** - but a set of time-varying allocation rules, of the form $\{\underline{\boldsymbol{w}}^\star(t)\}_{t \leq T}$. Indeed, the objective and constraint criteria do not penalize strategy that periodically violates then over-satisfies the constraint as long as it remains met over the whole trajectory. In contrast, an optimal stationary solution, denoted by $\underline{\boldsymbol{w}}^\star$, would ensure uniform performance and constraint satisfaction over the trajectory. This stability is crucial in applications where constraint satisfaction is monitored over sliding windows, or where oscillatory behavior must be strictly avoided, such as in clinical or medical trials. Interestingly, an optimal stationary policy can always be derived from an optimal time-varying one by leveraging the linearity of the problem and the use of expectations, constructing $\underline{\boldsymbol{w}}^\star = \frac{1}{T}\mathbb{E}\left[\sum_{t=1}^T \underline{\boldsymbol{w}}^\star(t)\right]$.

We focus from now on the stationary solution of the planning problem, which can be reformulated as solution of the linear program:

$$\mathbf{LP}(\underline{\boldsymbol{\mu}}^{obj}, \underline{\boldsymbol{\mu}}^{cons}): \max_{\underline{\boldsymbol{w}}} \quad f(\underline{\boldsymbol{\mu}}^{obj}, \underline{\boldsymbol{w}}) \qquad \text{with} \quad f(\underline{\boldsymbol{\mu}}, \underline{\boldsymbol{w}}) = \sum_{c \in \mathcal{C}} p_c \boldsymbol{\mu}_c^\top \boldsymbol{w}_c$$

$$\text{s. t.} \quad g_k(\underline{\boldsymbol{\mu}}^{cons}, \underline{\boldsymbol{w}}) \geq \lambda_k, \quad \forall k \in \mathcal{K}, \qquad g_k(\underline{\boldsymbol{\mu}}, \underline{\boldsymbol{w}}) = \sum_{c \in \mathcal{C}} p_c \boldsymbol{\mu}_c^\top \boldsymbol{e}_{kk} \boldsymbol{w}_c$$

$$h_k(c, \underline{\boldsymbol{w}}) \geq 0, \quad \forall (k,c) \in \mathcal{J}, \qquad h_k(c, \underline{\boldsymbol{w}}) = \boldsymbol{e}_k^\top \boldsymbol{w}_c, \quad q_c(\underline{\boldsymbol{w}}) = \mathbf{1}^\top \boldsymbol{w}_c$$

$$q_c(\underline{\boldsymbol{w}}) = 1, \quad \forall c \in \mathcal{C},$$

Namely, $f$ denotes the expected total revenue; $g_k$ represents the expected aggregated revenue of arm $k$ over all contexts; $h_k(c, \underline{w}) = w_{k,c}$ is the probability of selecting arm $k$ in context $c$; and $q_c$ is the $\ell_1$-norm of $\boldsymbol{w}_c$, used to ensure that the vector lies in the $K$-dimensional probability simplex. Hence, the optimal stationary allocation $\underline{w}^\star$ is the solution to $\mathbf{LP}(\underline{\mu}, \underline{\mu})$ [1] and sampling arm $k$ according to the optimal weights $\boldsymbol{w}_c^\star$ upon observing context $c$ yields the optimal strategy of interest for **OBJ**.

**Learning Problem.** At each round $t$, the learner utilizes the available information $\mathcal{H}_{t-1} \cup c_t$, and selects an arm according to learned allocations $\underline{w}(t)$. To evaluate the performance of the learner's algorithm, two key metrics are considered: the performance regret and the constraint violation.

**Definition 1.** *The cumulative regret $\mathcal{R}_T$ and the cumulative constraint violation $\mathcal{V}_T$ are respectively defined as:*

$$\mathcal{R}_T = \sum_{t=1}^{T} \left( f(\underline{\mu}, \underline{w}^\star) - f(\underline{\mu}, \underline{w}(t)) \right)_+, \quad \mathcal{V}_T = \sum_{t=1}^{T} \sum_{k \in \mathcal{K}} \left( \lambda_k - g_k(\underline{\mu}, \underline{w}(t)) \right)_+.$$

The positive part accounts only for non-negative deviations and its role is twofold. First, it favors stable long-term behavior and prevents convergence to optimal time-varying allocation. Second, it penalizes strategies that oscillate during the learning between being overly conservative and severely violating the constraints to ease the exploration, hence favoring a tracking of $\underline{w}^\star$ in a round wise stable manner.

## 2.2 ASSUMPTIONS

We rely on the following standard assumptions, which concern the stochastic nature of the setting and the feasibility of the associated planning problem.

**Assumption 1** (**Sub-Gaussian rewards**). *The reward distributions are conditionally 1-sub-Gaussian:*

$$\forall b \in \mathbb{R}, \quad \mathbb{E}\left[ \exp\left( b\left( r_t - \mathbb{E}[r_t \mid (c_t, k_t)] \right) \right) \mid (c_t, k_t) \right] \leq \exp\left( \frac{b^2}{2} \right).$$

**Assumption 2** (**Known Contexts Probabilities**). *The contexts probabilities $\{p_c\}_{c \in \mathcal{C}}$ are known.*

We adopt the same context prior assumption as Guo et al. (2025), which is here mild and used primarily for expository simplicity. In Section 7, we discuss a straightforward relaxation of this assumption.

**Assumption 3** (**Feasibility and Non-degeneracy**). *The optimization problem $\mathbf{LP}(\underline{\mu}, \underline{\mu})$ is feasible and non-degenerate.*

**Assumption 4** (**Strict Feasibility and Non-degeneracy**). *The optimization problem $\mathbf{LP}(\underline{\mu}, \underline{\mu})$ is strictly feasible and non-degenerate.*

Assumption 4 is specifically required for **OPLP**, with the corresponding feasibility margin quantified by $\gamma^\star$. In contrast, **OLP** only requires Assumption 3, and its guarantees are independent of the slack parameter $\gamma^\star$.

**Definition 2** (**Feasibility Margin**).

$$\gamma^\star := \max\left\{ s \in \mathbb{R}_+^\star \mid \Phi(s) \neq \emptyset \right\}, \quad \text{where} \quad \Phi(s) := \left\{ \underline{w} \in \pi_K^{|\mathcal{C}|} \mid \forall k \in \mathcal{K}, \; g_k(\underline{\mu}, \underline{w}) \geq \lambda_k + s \right\}.$$

In addition, we quantify the sensitivity of the optimal performance w.r.t. uniform constraint perturbations through a problem dependent constant $S_{\gamma^\star}$, intrinsically linked to the feasibility margin. We prove in Appendix B, Prop. B.1 that $S_{\gamma^\star} < \infty$ for any **MAB-ARC** instance satisfying Assumption 3.

**Definition 3** (**Performance Sensitivity Coefficient**).

$$S_{\gamma^\star} := \min\{ S \in \mathbb{R}_+ : \forall 0 \leq s_1 < s_2 \leq \gamma^\star, \; \max_{\underline{w} \in \Phi(s_1)} f(\underline{\mu}, \underline{w}) - \max_{\underline{w} \in \Phi(s_2)} f(\underline{\mu}, \underline{w}) \leq S(s_2 - s_1) \}.$$

---

[1] For brevity, we adopt the shorthand: $\underline{w}^\star = \arg\max_{\underline{w}} \mathbf{LP}(\underline{\mu}, \underline{\mu})$.

## 3 ORACLE-GUIDED BEHAVIOR AND OPTIMALITY CHARACTERIZATION

**Proper Tracking of the Optimal Planning.** The problem formulation yields a linear program (**LP**) optimizing allocations over the probability simplex, whose solution lies at a vertex determined by binding constraints (Boyd and Vandenberghe, 2004; Nesterov, 2014). The characterization of the optimal solution can thus be decomposed into identifying the optimal active set of constraints $\mathcal{I}^\star$ for $\mathbf{LP}(\underline{\boldsymbol{\mu}}, \underline{\boldsymbol{\mu}})$, then computing the best allocation that saturates these constraints. More precisely:

(i) If $k \in \mathcal{I}^\star \cap \mathcal{K}$, then $g_k(\underline{\boldsymbol{\mu}}, \underline{\boldsymbol{w}}^\star) = \lambda_k$, indicating that arm $k$ exactly attains its minimum required revenue.

(ii) If $(k, c) \in \mathcal{I}^\star \cap \mathcal{J}$, then $h_k(c, \underline{\boldsymbol{w}}^\star) = 0$, which implies that arm $k$ is never selected in context $c$, i.e., $w_{k,c}^\star = 0$.

Consequently, the oracle effectively solves the optimization problem $\mathbf{OPT}(\underline{\boldsymbol{\mu}}, \underline{\boldsymbol{\mu}}, \mathcal{I}^\star)$, defined as:

$$\mathbf{OPT}(\underline{\boldsymbol{\mu}}^{obj}, \underline{\boldsymbol{\mu}}^{cons}, \mathcal{I}) : \quad \underset{\underline{\boldsymbol{w}}}{\text{maximize}} \quad f(\underline{\boldsymbol{\mu}}^{obj}, \underline{\boldsymbol{w}})$$

$$\text{subject to} \quad g_k(\underline{\boldsymbol{\mu}}^{cons}, \underline{\boldsymbol{w}}) = \lambda_k, \quad \forall k \in \mathcal{K} \cap \mathcal{I},$$

$$h_k(c, \underline{\boldsymbol{w}}) = 0, \quad \forall (k, c) \in \mathcal{J} \cap \mathcal{I},$$

$$q_c(\underline{\boldsymbol{w}}) = 1, \quad \forall c \in \mathcal{C}.$$

Directly quantifying the sub-optimality of an allocation $\underline{\boldsymbol{w}}$ w.r.t. $\underline{\boldsymbol{w}}^\star$ is challenging due to the linear nature of the allocation problem. Yet, the introduction of the intermediate quantity $\mathcal{I}$ allows us to retrieve a notion of gap, similarly to standard MAB problem. In what follow, we define $\rho(\mathcal{I})$ which quantifies the sub-optimality of a candidate set $\mathcal{I}$. $\rho(\mathcal{I})$ will play a key role in showing that $\mathcal{I}^\star$ can be quickly identified by a learning strategy.

**Optimality Characterization.** One form of sub-optimality arises from infeasibility—that is, the absence of any allocation that satisfies the constraints in $\mathcal{I}$. We formalize this as follows.

**Definition 4 (Feasibility Gap).** *For any set $\mathcal{I}$, define:* $s(\mathcal{I}) = \min \left\{ s \geq 0 \; : \; \psi(s, \mathcal{I}) \neq \emptyset \right\}$, *where:*

$$\psi(s, \mathcal{I}) = \left\{ \underline{\boldsymbol{w}} \in \pi_K^{|\mathcal{C}|} \; : \; \begin{array}{ll} \forall k \in \mathcal{K}, & g_k(\underline{\boldsymbol{\mu}}, \underline{\boldsymbol{w}}) \geq \lambda_k - s, \\ \forall k \in \mathcal{K} \cap \mathcal{I}, & g_k(\underline{\boldsymbol{\mu}}, \underline{\boldsymbol{w}}) \leq \lambda_k + s, \\ \forall (k, c) \in \mathcal{J} \cap \mathcal{I}, & h_k(c, \underline{\boldsymbol{w}}) = 0 \end{array} \right\}.$$

The quantity $s(\mathcal{I})$ captures the minimum slack required to make the revenue constraints feasible while saturating - up to some margin - the arms in $\mathcal{K} \cap \mathcal{I}$ and satisfying the allocation sparsity prescribed by $\mathcal{J} \cap \mathcal{I}$ (i.e, the set of $(k, c)$ s.t. $w_{k,c} = 0$).

Beyond feasibility, we also quantify sub-optimality from a performance standpoint.

**Definition 5 (Performance Gap).** *For any candidate set $\mathcal{I}$, we define the performance sensitivity $\mathcal{L}(\mathcal{I})$ and the performance gap $\mathcal{P}(\mathcal{I})$ as:*

$$\mathcal{L}(\mathcal{I}) := \min\{\mathcal{L} \in \mathbb{R}_+ : \forall s(\mathcal{I}) \leq s_1 < s_2, \; \max_{\underline{\boldsymbol{w}} \in \psi(s_2, \mathcal{I})} f(\underline{\boldsymbol{\mu}}, \underline{\boldsymbol{w}}) - \max_{\underline{\boldsymbol{w}} \in \psi(s_1, \mathcal{I})} f(\underline{\boldsymbol{\mu}}, \underline{\boldsymbol{w}}) \leq \mathcal{L}(s_2 - s_1)\},$$

$$\mathcal{P}(\mathcal{I}) := \left( f(\underline{\boldsymbol{\mu}}, \underline{\boldsymbol{w}}^\star) - \max_{\underline{\boldsymbol{w}} \in \psi(s(\mathcal{I}), \mathcal{I})} f(\underline{\boldsymbol{\mu}}, \underline{\boldsymbol{w}}) \right) / \left( \max(1, S_{\gamma^\star}) + \mathcal{L}(\mathcal{I}) \right).$$

The denominator of $\mathcal{P}(\mathcal{I})$, beyond its technical role in the proof, can be interpreted as a scaling factor that reflects both the geometry of the candidate set $\mathcal{I}$ and the problem's sensitivity to perturbations. Proposition B.2 in Appendix B shows that, for any candidate $\mathcal{I}$, $\mathcal{L}(\mathcal{I})$ is finite . Combining feasibility and performance considerations, we define the overall sub-optimality gap as follows:

**Definition 6 (Sub-optimality Gap).** *For a candidate set $\mathcal{I}$, the sub-optimality gap is defined as $\rho(\mathcal{I}) := \max (s(\mathcal{I}), \mathcal{P}(\mathcal{I}))$, with the worst-case sub-optimality given by $\rho^\star := \min_{\mathcal{I}:\mathcal{I} \neq \mathcal{I}^\star} \rho(\mathcal{I})$.*

This characterization enables distinguishing optimal from suboptimal sets of saturated constraints. Lemma 3.1 formalizes this, showing that $\rho(\mathcal{I})$ plays a role analogous to the gap in classical MAB. The proof is deferred to Appendix B.1.

**Lemma 3.1 (Suboptimality Characterization).** *Under Assumption 3,* $\quad \rho(\mathcal{I}^\star) = 0 \quad$ *and* $\quad \rho^\star > 0$.

## 4 ALGORITHMS

The learner's objective is to find the optimal allocation $\underline{w}^\star$ without prior knowledge of the true parameters $\underline{\mu}$. While estimates can be constructed from data, the agent must carefully trade-off between exploration and exploitation. We summarize in this section the confidence set construction and our proposed strategies that focus either on performance or on constraints satisfaction.

**Confidence Set.** The unknown parameter $\underline{\mu}$ can be estimated online from past interactions with a standard empirical mean, formally given by:

$$\hat{\mu}_{k,c}(t) = \frac{1}{n_{k,c}(t)}\sum_{s=1}^{t} r_s \mathbb{1}_{[k_s=k,\,c_s=c]}, \text{ where } n_{k,c}(t) = \sum_{s=1}^{t}\mathbb{1}_{[k_s=k,\,c_s=c]}. \tag{1}$$

Further, the concentration of the empirical estimator is prescribed by the set

$$\mathcal{S}_t(\underline{\hat{\mu}}(t),\delta) = \left\{\underline{\mu} : \forall (k,c),\ |\hat{\mu}_{k,c}(t) - \mu_{k,c}| \leq \epsilon_{k,c}(t)\right\}, \text{ with } \epsilon_{k,c}(t) = \sqrt{\frac{2\log\left(\frac{2\kappa}{\delta}\right)}{n_{k,c}(t-1)}}. \tag{2}$$

Proposition 4.1 ensures that $\mathcal{S}_t$ is a valid confidence set for $\underline{\mu}$ and provides prediction error bounds on the performance and constraints violation. The proof is deferred to Appendix C.

**Proposition 4.1 (Confidence set).** *Under Assumption 1, let $\underline{\hat{\mu}}(t)$ and $\mathcal{S}_t$ defined in Eq. 1 and 2, then:*

*(i)* $\forall t \geq 1$, $\mathbb{P}\left(\underline{\mu} \in \mathcal{S}_t(\underline{\hat{\mu}}(t),\delta)\right) \geq 1-\delta$,

*(ii)* $\forall t \geq 1$, *w.p. at least* $1-\delta$, *for any* $\underline{w} \in \pi_K^{|\mathcal{C}|}$, $\underline{\tilde{\mu}}(t) \in \mathcal{S}_t(\underline{\hat{\mu}}(t),\delta)$ *and* $k \in \mathcal{K}$,

$$\left|g_k(\underline{\tilde{\mu}}(t),\underline{w}) - g_k(\underline{\mu},\underline{w})\right| \leq \rho_k(\underline{\epsilon}(t),\underline{w}), \quad \text{where} \quad \rho_k(\underline{\epsilon}(t),\underline{w}) = \sum_{c \in \mathcal{C}} 2\epsilon_{k,c}(t)w_{k,c}(t),$$

$$\left|f(\underline{\tilde{\mu}}(t),\underline{w}) - f(\underline{\mu},\underline{w})\right| \leq \rho(\underline{\epsilon}(t),\underline{w}), \qquad \rho(\underline{\epsilon}(t),\underline{w}) = \sum_{k \in [K]} \rho_k(\underline{\epsilon}(t),\underline{w}).$$

Equipped with the confidence set construction, we define the upper and lower confidence bounds as $\underline{\mathbf{UCB}}(t) = \underline{\hat{\mu}}(t) + \underline{\epsilon}(t)$ and $\underline{\mathbf{LCB}}(t) = \underline{\hat{\mu}}(t) - \underline{\epsilon}(t)$, both of which belong to $\mathcal{S}_t(\underline{\hat{\mu}}(t),\delta)$.

---

**OLP:** Optimistic Linear Programming

1 **Inputs:** $\{\lambda_k\}_{k\in\{1,\dots,K\}}, \{p_c\}_{c\in\mathcal{C}}$
2 **for** $t = 1,\dots,T$ **do**
3    Observe context $c_t$
4    Set $\delta \leftarrow 1/t$
5    $\underline{w}(t) = \underset{\underline{w}\in\pi_K^{|\mathcal{C}|}}{\mathrm{argmax}}\ \mathbf{LP}(\underline{\mathbf{UCB}}(t), \underline{\mathbf{UCB}}(t))$
6    Sample arm $k_t \sim \underline{w}_{c_t}(t)$
7    Receive reward $r_t \sim \mu_{k_t,c_t}$
8    Update $\underline{n}(t), \underline{\hat{\mu}}(t), \underline{\epsilon}(t)$
9    Update history $\mathcal{H}_t = \mathcal{H}_{t-1} \cup \{c_t, k_t, r_t\}$
10 **end**

---

**OPLP:** Optimistic-Pessimistic Linear Programming

1 **Inputs:** $\{\lambda_k\}_{k\in\{1,\dots,K\}}, \{p_c\}_{c\in\mathcal{C}}$
2 **for** $t = 1,\dots,T$ **do**
3    Observe context $c_t$
4    Set $\delta \leftarrow 1/t$
5    **if** $LP(\underline{\mathbf{UCB}}(t), \underline{\mathbf{LCB}}(t))$ *is feasible* **then**
6      $\underline{w}(t) = \underset{\underline{w}\in\pi_K^{|\mathcal{C}|}}{\mathrm{argmax}}\ \mathbf{LP}(\underline{\mathbf{UCB}}(t), \underline{\mathbf{LCB}}(t))$
7    **end**
8    **else**
9      $\underline{w}(t) = \underset{\underline{w}\in\pi_K^{|\mathcal{C}|}}{\mathrm{argmax}}\ \mathbf{LP}(\underline{\mathbf{UCB}}(t), \underline{\mathbf{UCB}}(t))$
10    **end**
11    Sample arm $k_t \sim \underline{w}_{c_t}(t)$
12    Receive reward $r_t \sim \mu_{k_t,c_t}$
13    Update $\underline{n}(t), \underline{\hat{\mu}}(t), \underline{\epsilon}(t)$
14    Update history $\mathcal{H}_t = \mathcal{H}_{t-1} \cup \{c_t, k_t, r_t\}$
15 **end**

---

We propose two algorithms that focus either on the performance or on the constraint violation. **OLP** adopts an optimistic approach by solving the underlying **LP** problem using $\underline{\mathbf{UCB}}(t)$ as a parameter for both the objective function and the constraints. Under Asm. 3, the inner maximization problem remains feasible at all times.

On the other hand, **OPLP** proposes an asymmetric estimation strategy that leverages an optimistic estimate $\underline{\mathbf{UCB}}(t)$ for the objective and a pessimistic estimate $\underline{\mathbf{LCB}}(t)$ for the constraints parameters. In contrast with **OLP**, the inner maximization problem may not always be feasible. In such cases, a fallback procedure based on a doubly optimistic approach is used instead.

## 5 MAIN RESULTS

### 5.1 ALGORITHM GUARANTEES

The following theorems provide regret and constraint violation guarantees for **OLP** and **OPLP**, highlighting their respective focus on reward performance and constraint satisfaction. The proof sketches are presented in Appendix D, while the detailed proofs are deferred to Appendix E.

**Theorem 5.1** (Upper bounds for **OLP**). *Under Assumptions 1, 2 and 3, the performance and constraint regret of **OLP** satisfy:*

$$\mathbb{E}[\mathcal{R}_T] \leq \mathcal{O}\left(\frac{\log{(T)}^2}{\rho^\star}\right),$$

$$\mathbb{E}[\mathcal{V}_T] \leq \mathcal{O}\left(\frac{\log{(T)}^2}{\rho^\star} + \sqrt{|\mathcal{K} \cap \mathcal{I}^\star| \log{(T)} \, T}\right).$$

**Theorem 5.2** (Upper bounds for **OPLP**). *Under Assumptions 1, 2 and 4, the performance and constraint regret of **OPLP** satisfy:*

$$\mathbb{E}[\mathcal{R}_T] \leq \mathcal{O}\left(\left(\frac{1}{\gamma^{\star 2}} + \frac{1}{\rho^{\star 2}}\right)\log{(T)}^2 + \sqrt{|\mathcal{K} \cap \mathcal{I}^\star| \log{(T)} \, T}\right),$$

$$\mathbb{E}[\mathcal{V}_T] \leq \mathcal{O}\left(\frac{\lambda}{\gamma^{\star 2}}\log{(T)}^2\right), \text{ where } \lambda = \sum_{k \in \mathcal{K}} \lambda_k.$$

**Discussion.** **OLP** and **OPLP** enjoy regret guarantees that stand at two different points in the performance/constraint violation Pareto front. **OLP** prioritizes performance, achieving polylogarithmic regret, but may incur constraint violations as large as $\mathcal{O}(\sqrt{T})$. Interestingly, its bounds adapt to the number of arms that saturate their minimum reward constraints. In particular, when no arm saturates its constraint [2] (i.e $|\mathcal{K} \cap \mathcal{I}^\star| = 0$), we recover polylogarithmic guarantees for both regret and constraint violations. In contrast, **OPLP** emphasizes constraint satisfaction at the cost of performance. Its theoretical guarantees depend on a richer set of problem-dependent constants. In Theorems 5.1 and 5.2, $\rho^\star$ characterizes the speed at which each algorithm converges to the optimal set of saturated constraints, i.e., the point at which $\mathcal{I}_t = \mathcal{I}^\star$. The constant $\gamma^\star$ - which appears only in **OPLP** - arises from its intrinsic phased structure and quantifies how fast **LCB** becomes feasible. The pessimistic strategy ensures a more conservative treatment of constraints by incorporating safety margins. However, this comes at the expense of performance, as a portion of the allocation budget is diverted from non-saturating (typically high-reward) arms to those saturating their constraints leading to $\sqrt{T}$ loss in performance.

### 5.2 LOWER BOUND

In line with previous works in the non-contextual setting, **OLP** and **OPLP** enjoys a cumulated guarantee on $\mathcal{R} + \mathcal{V}$ of order $\sqrt{T}$. On the other hand, the performance (resp. constraint violation) regret bound for **OLP** (resp. **OPLP**) is only logarithmic, in contrast with the no-regret (constant) guarantee of (Baudry et al., 2024). We propose in this section a lower bound which stresses this is not due to algorithmic design or analysis weaknesses but structural to the **MAB-ARC** setting. In particular, this refutes the *free exploration* property leveraged in prior work as soon as $|\mathcal{C}| > 1$ and $K > 2$.

Let $\boldsymbol{\nu} = (\underline{\boldsymbol{\mu}}, \underline{\boldsymbol{\lambda}}, \{p_c\}_{c \in \mathcal{C}})$ represent a generic **MAB-ARC** instance, and denote by $\mathcal{R}_{\boldsymbol{\nu},\pi}(T)$ and $\mathcal{V}_{\boldsymbol{\nu},\pi}(T)$ the performance and constraint regret under policy $\pi$ on instance $\boldsymbol{\nu}$. We consider a nominal instance $\boldsymbol{\nu}^{(0)}$ with $K = 3$ arms and $|\mathcal{C}| = 3$ contexts as well as a set of nearby instances $\Upsilon(\boldsymbol{\nu}^{(0)}, \varepsilon)$ defined in Table. 1 and Eq. 3 respectively.

Table 1: Nominal instance $\boldsymbol{\nu}^{(0)}$.

| $k$ | $p_c \, \mu_{k,c}$ | | | $\lambda$ |
| --- | --- | --- | --- | --- |
| | $c = 1$ | $c = 2$ | $c = 3$ | |
| 1 | $\mu_{1,1} = 3$ | $\mu_{1,2} = 1$ | $\mu_{1,3} = 1$ | $1$ |
| 2 | $\mu_{2,1} = 0$ | $\mu_{2,2} = \frac{1}{2}$ | $\mu_{2,3} = 0$ | $\frac{1}{4}$ |
| 3 | $\mu_{3,1} = 0$ | $\mu_{3,2} = 0$ | $\mu_{3,3} = 2$ | $1$ |

$$\Upsilon(\boldsymbol{\nu}^{(0)}, \varepsilon) = \left\{\boldsymbol{\nu} = (\underline{\boldsymbol{\mu}}, \underline{\boldsymbol{\lambda}}^{(0)}, \{p_c^{(0)}\}_c) : p_2 \left|\mu_{2,2} - \mu_{2,2}^{(0)}\right| \leq \frac{\varepsilon}{2}, \text{ otherwise } \mu_{k,c} = \mu_{k,c}^{(0)}\right\} \quad (3)$$

---

[2] This occurs when the revenue constraints are small compared to the optimal performance of each arm.

**Theorem 5.3** (**Lower Bound**). *Let $\boldsymbol{\nu}^{(0)}$ and $\Upsilon(\boldsymbol{\nu}^{(0)}, \varepsilon)$ defined in Table 1 and Eq. 3, then:*

*(i) For $T \geq 16$, there exists $\varepsilon_T$ small enough such that:*

$$\min_{\pi} \max_{\boldsymbol{\nu} \in \Upsilon(\boldsymbol{\nu}^{(0)}, \varepsilon_T)} \mathbb{E}\left[\mathcal{R}_{\boldsymbol{\nu}, \pi}(T) + \mathcal{V}_{\boldsymbol{\nu}, \pi}(T)\right] = \Omega\left(\sqrt{T}\right).$$

*(ii) For any consistent policy $\pi$, $\exists\, T_0 \geq 0$ s.t. $\forall\, T \geq T_0$, $\quad \mathbb{E}\left[\mathcal{R}_{\boldsymbol{\nu}^{(0)}, \pi}(T)\right] = \Omega\left(\log T\right).$*

**Discussion.** Theorem 5.3 establishes two distinct results (the proof is deferred in Appendix F.1). The first assertion proposes a locally minimax lower bound around the nominal instance $\boldsymbol{\nu}^{(0)}$ and confirms that no strategy can enjoy a cumulated regret $\mathcal{R} + \mathcal{V}$ uniformly better than $\sqrt{T}$ in a neighborhood of $\boldsymbol{\nu}^{(0)}$. There always exists a nearby alternative which suffers from large performance ($\mathcal{R}$) or constraint violation ($\mathcal{V}$) regret. Such result offers a finite time counterpart to the asymptotic lower bound of (Baudry et al., 2024) extended to the contextual setting but limited to the instance $\boldsymbol{\nu}^{(0)}$.

Theorem 5.3 $(i)$ indicates that both **OLP** and **OPLP** offer the correct $\sqrt{T}$ dependency (but for logarithmic factor) for the overall regret $\mathcal{R} + \mathcal{V}$ but leaves open the question of whether the pair $(\mathcal{R}, \mathcal{V})$ is optimally positioned in the performance/constraint violation Pareto front. Indeed, in the non-contextual setting, constant performance regret and $\sqrt{T}$ constraint violation is attainable.

The second assertion $(ii)$ rules out this possibility in the contextual setting and shows that no policy can offer better guarantees than $\sqrt{T}$ constraint violation and $\log(T)$ performance regret for the instance $\boldsymbol{\nu}^{(0)}$. This demonstrates the near-optimality of **OLP** with respect to $T$ but more importantly refutes the possibility of *free exploration* in the contextual setting. This is in shark contrast with the non-contextual setting where all arms are sampled linearly with $T$, a property heavily exploited in (Baudry et al., 2024). In **MAB-ARC**, optimal allocations may assign zero probability to certain arms, meaning that no natural exploration occurs, which reinstates the exploration-exploitation trade-off. Notice that while $\boldsymbol{\nu}^{(0)}$ exhibits such optimal allocation structure, the following lemma ensures this is shared among a large family of instances. The proof of Lemma 5.1 is deferred to Appendix F.2.

**Lemma 5.1.** *For any **MAB-ARC** instance such that $K > 2$ and $|\mathcal{C}| > 1$, there exists at least one pair $(k, c) \in \mathcal{J}$ for which the optimal allocation satisfies $w_{k,c}^{\star} = 0$.*

**Numerical Illustrations.** For completeness, we conduct numerical evaluations on synthetic data to concretely illustrate the concept of $\mathcal{I}^{\star}$ and to compare our algorithms with **Optimistic**[3] from Guo et al. (2025), as well as with **DOC** and **SPOC** from Baudry et al. (2024). We further examine the sensitivity of the **OPLP** algorithm with respect to variations in $\gamma^{\star}$. Complete experimental details and results are provided in Appendix G.

## 6 COMPUTATIONAL COMPLEXITY

**Saving Computation:** To reduce computational cost, we employ a lazy update scheme. Rather than solving the LP in every round, the policy is recomputed only when the number of pulls for any arm–context pair has doubled since the last update. Between these events, the confidence bounds and resulting policy remain unchanged. This ensures that the LP is solved at most $O(\log T)$ times over the horizon $T$, yielding substantial computational savings while preserving the original regret guarantees up to constant factors. Detailed implementation and theoretical justification are provided in Appendix H.

**Computational Overhead and Practical Deployment:** Our method requires solving a linear program frequently, resulting in a higher per-iteration cost than the gradient-based approach of Guo et al. (2025). However, their computational efficiency is achieved at the expense of weaker theoretical guarantees. Consequently, a practitioner faces a clear trade-off: stronger theoretical guarantees (ours) versus greater computational efficiency Guo et al. (2025). The choice between them depends on the specific priorities of the application. From a practical standpoint, the computational burden of our approach can be mitigated through several techniques. Modern LP solvers reduce overhead via *presolve* to simplify problems and *warm-starting* to reuse prior solutions of similar LP. Furthermore, solver tolerance can be relaxed to match the statistical confidence width of reward estimates, avoiding unnecessary precision without impacting practical performance.

## 7 LIMITATIONS AND FUTURE WORK

**Context Prior:** Assumption 2 is mild and can be relaxed through appropriate empirical estimation. The key insight is that the estimation target shifts from the conditional reward $\mu_{k,c}$ to the product $\mu'_{k,c} = p_c \mu_{k,c}$, which integrates the context probability. Once optimistic and pessimistic estimators for $\mu'_{k,c}$ are constructed, they integrate directly into the **OLP** and **OPLP** frameworks. However, effective estimation in this case requires additional smoothness assumptions on the context distribution. We anticipate that similar theoretical guarantees can be obtained through a direct adaptation of our analytical techniques.

**Infeasibility Issue:** If Assumption 3 is violated, infeasibility of **LP**$(\mu, \mu)$ is detected with high probability when the optimistic counterpart **LP(UCB, UCB)** is infeasible. While no unique corrective procedure is prescribed, a standard recourse is to relax the problem to find the closest feasible approximation. A possible strategy is to iteratively scale the constraint thresholds until feasibility is restored, for instance, via the update $\lambda_k \leftarrow \alpha \lambda_k$, where $\alpha \in (0,1)$ is a judiciously chosen scaling parameter.

**Greedy Policy:** Beyond the optimistic and pessimistic strategies discussed in this paper, the greedy policy is well-known but inefficient in the contextual setting. In the single-context case Baudry et al. (2024), it achieves sublinear regret as constraints enforce exploration: satisfying per-arm revenue constraints requires playing all arms, improving estimates over time. In contrast, in the multi-context setting, constraints do not inherently induce exploration—a counterexample illustrating this is provided in the Appendix I.

**Future Work:** Extending our framework to infinite context or action sets is a compelling direction for future work. Conceptually, the **OLP** and **OPLP** approaches are based on constructing optimistic or pessimistic reward estimates and then solving a linear program defined by expectations with respect to the context distribution. Technically, efficient reward estimation in infinite spaces requires structural assumptions to ensure tractability. A standard approach is to posit a linear model of the form $\mu_{k,c} = \langle \theta^\star, \phi(k,c) \rangle$, where $\phi$ is a known feature mapping. Using established results from linear bandit theory, one can then derive efficient optimistic and pessimistic reward estimates for all context–action pairs. Alternatively, kernelized methods could be employed by assuming the mean reward function lies in a reproducing kernel Hilbert space (RKHS), which also enables the construction of practical confidence estimators. Once such estimators are available and the corresponding LP can be solved, we anticipate that a theoretical analysis analogous to ours could be developed. However, this would also necessitate research into discretization techniques or heuristics for solving the resulting optimization problem efficiently, accompanied by a formal analysis of the induced approximation errors. While highly interesting, this comprehensive extension falls outside the scope of the present paper on multi-armed bandits, as it constitutes a separate research centered on linear and kernelized bandits.

## CONCLUSION

We introduced a novel contextual bandit problem with minimum aggregated reward constraints, along with analytical tools tailored to the structure of this constrained optimization problem. We proposed two algorithms that explore different regions of the Pareto frontier—one favoring performance, the other emphasizing constraint satisfaction. Our upper bound analysis highlights the adaptability of the proposed approach across regimes with both saturating and non-saturating constraints, outperforming standard linear bandit models that rely on self-normalized concentration inequalities and fail to capture the fine structure of the problem. We also established a lower bound that confirms the near optimality of our upper bounds and challenges the previously leveraged notion of *free exploration* in the non-contextual setting. While our primary focus is on guaranteeing a minimum aggregated revenue per arm, the algorithmic and analytical framework generalizes naturally to broader constraint structures, such as ensuring that the cumulative reward from a subset of arms exceeds a given threshold—a formulation relevant in generic monitoring problems.

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

APPENDICES

APPENDIX CONTENTS

# A USEFUL INEQUALITIES

**Theorem A.1** (Hoeffding's Inequality). *Let $X_1, \ldots, X_n$ be a sequence of independent 1-subgaussian random variables with mean $\mu$. Define $\hat{\mu} = \frac{1}{n} \sum_{i=1}^{n} X_i$. Then, for any $\epsilon > 0$, we have:*

$$\mathbb{P}\left(|\hat{\mu} - \mu| \geq \epsilon\right) \leq 2 \exp\left(-\frac{n\epsilon^2}{2}\right). \tag{4}$$

**Fact A.1.** *For sufficiently large $T$, the following inequality holds:*

$$\sum_{t=1}^{T} \sqrt{\frac{1}{t}} \leq 2[\sqrt{T} - 1] + 1 \leq \mathcal{O}\left(\sqrt{T}\right) \tag{5}$$

**Fact A.2.** *For sufficiently large $T$, the following inequality holds:*

$$\sum_{t=1}^{T} \frac{1}{t} \leq \log(T) + 1 \leq \mathcal{O}\left(\log(T)\right) \tag{6}$$

# B ORACLE BEHAVIOR

**Lemma B.1.** *Let $\underline{w}$ be the optimal solution of the problem $\textbf{OPT}(\underline{\mu}, \mu, \mathcal{I}^\star)$. The following property holds:*

$$\textit{If } \mathbb{1}_{[(k,c) \in \mathcal{J} \cap \mathcal{I}^\star]} = 0, \textit{ then } \mathbb{1}_{[k \in \mathcal{K} \cap \mathcal{I}^\star]} = 0 \Longrightarrow \mu_{k,c} = \|\mu_c\|_\infty.$$

In other words, for any arm $k$ in context $c$, if the optimal allocation probability $w_{k,c}$ is strictly positive ($w_{k,c} > 0$) and the arm's revenue $g_k(\underline{\mu}, \underline{w})$ exceeds its minimum constraint $\lambda_k$ (i.e., $g_k(\underline{\mu}, \underline{w}) > \lambda_k$), then arm $k$ must necessarily be the optimal arm for context $c$.

*Proof of Lemma B.1.* The proof is based in the dual analysis of the optimization problem. Recall:

$$\textbf{OPT}(\underline{\mu}, \underline{\mu}, \mathcal{I}) : \quad \begin{aligned} &\underset{\underline{w}}{\text{maximize}} && f(\underline{\mu}, \underline{w}) \\ &\text{subject to} && g_k(\underline{\mu}, \underline{w}) = \lambda_k, \quad \forall k \in \mathcal{K} \cap \mathcal{I}, \\ &&& h_k(c, \underline{w}) = 0, \quad \forall (k,c) \in \mathcal{J} \cap \mathcal{I}, \\ &&& q_c(\underline{w}) = 1, \quad \forall c \in \mathcal{C}. \end{aligned}$$

Let $\alpha$, $\beta$, and $\eta$ be dual vectors of dimensions $K$, $\kappa$, and $|\mathcal{C}|$, respectively. The Lagrangian function associated with the primal problem is:

$$\mathcal{L}(\underline{w}, \alpha, \beta, \eta) = f(\underline{\mu}, \underline{w}) + \sum_{k \in \mathcal{K} \cap \mathcal{I}^\star} \alpha_k \left(g_k(\underline{\mu}, \underline{w}) - \lambda_k\right) + \sum_{(k,c) \in \mathcal{J} \cap \mathcal{I}^\star} \beta_{kc} h_k(c, \underline{w}) + \sum_{c \in \mathcal{C}} \eta_c \left(q_c(\underline{w}) - 1\right)$$

$$= f(\underline{\mu}, \underline{w}) + \sum_{k \in \mathcal{K} \cap \mathcal{I}^\star} \alpha_k g_k(\underline{\mu}, \underline{w}) + \sum_{(k,c) \in \mathcal{J} \cap \mathcal{I}^\star} \beta_{kc} h_k(c, \underline{w}) + \sum_{c \in \mathcal{C}} \eta_c q_c(\underline{w}) - \sum_{k \in \mathcal{K} \cap \mathcal{I}^\star} \lambda_k \alpha_k - \sum_{c \in \mathcal{C}} \eta_c$$

$$= \sum_{c \in \mathcal{C}} \mu_c^\top w_c + \sum_{k \in \mathcal{K} \cap \mathcal{I}^\star} \alpha_k \sum_{c \in \mathcal{C}} \mu_c^\top e_{kk} w_c + \sum_{(k,c) \in \mathcal{J} \cap \mathcal{I}^\star} \beta_{kc} e_k^\top w_c + \sum_{c \in \mathcal{C}} \eta_c \mathbf{1}^\top w_c - \sum_{k \in \mathcal{K} \cap \mathcal{I}^\star} \lambda_k \alpha_k - \sum_{c \in \mathcal{C}} \eta_c$$

$$= \sum_{c \in \mathcal{C}} \mu_c^\top w_c + \sum_{c \in \mathcal{C}} \mu_c^\top \left(\sum_{k=1}^{K} \alpha_k \mathbb{1}_{[k \in \mathcal{K} \cap \mathcal{I}^\star]} e_{kk}\right) w_c + \sum_{c \in \mathcal{C}} \left(\sum_{k=1}^{K} \mathbb{1}_{[(k,c) \in \mathcal{J} \cap \mathcal{I}^\star]} \beta_{kc} e_k^\top\right) w_c + \sum_{c \in \mathcal{C}} \eta_c \mathbf{1}^\top w_c$$

$$- \sum_{k \in \mathcal{K} \cap \mathcal{I}^\star} \lambda_k \alpha_k - \sum_{c \in \mathcal{C}} \eta_c$$

$$= \sum_{c \in \mathcal{C}} \underbrace{\left(\mu_c + \left(\sum_{k=1}^{K} \alpha_k \mathbb{1}_{[k \in \mathcal{K} \cap \mathcal{I}^\star]} e_{kk}\right) \mu_c + \sum_{k=1}^{K} \beta_{kc} \mathbb{1}_{[(k,c) \in \mathcal{J} \cap \mathcal{I}^\star]} e_k + \eta_c \mathbf{1}\right)}_{a_c}^\top w_c - \sum_{k \in \mathcal{K} \cap \mathcal{I}^\star} \lambda_k \alpha_k - \sum_{c \in \mathcal{C}} \eta_c$$

Given that the primal problem is feasible and finite, the dual problem is also feasible and finite. This implies the following condition:

$$\boldsymbol{a}_c = 0, \quad \forall c \in \mathcal{C}$$
$$\Leftrightarrow \mu_{k,c} + \alpha_k \mathbb{1}_{[k \in \mathcal{K} \cap \mathcal{I}^\star]} \mu_{k,c} + \beta_{kc} \mathbb{1}_{[(k,c) \in \mathcal{J} \cap \mathcal{I}^\star]} + \eta_c = 0, \quad \forall k \in \mathcal{K}, c \in \mathcal{C} \tag{7}$$

Let $k, c$ such that $\mathbb{1}_{[(k,c) \in \mathcal{J} \cap \mathcal{I}^\star]} = 0$ meaning that $\boldsymbol{w}_{k,c} \neq 0$. Hence, equation (7) becomes:

$$\mu_{k,c} + \alpha_k \mathbb{1}_{[k \in \mathcal{K} \cap \mathcal{I}^\star]} \mu_{k,c} = -\eta_c$$

From this, we deduce:

$$\mathbb{1}_{[k \in \mathcal{K} \cap \mathcal{I}^\star]} = 0 \implies \mu_{k,c} = -\eta_c.$$

From strong duality we have : $\mathbf{OPT}(\underline{\boldsymbol{\mu}}, \boldsymbol{\mu}, \mathcal{I}) = -\sum_{k \in \mathcal{K} \cap \mathcal{I}^\star} \lambda_k \alpha_k - \sum_{c \in \mathcal{C}} \eta_c$, hence to maximize the performance, $-\eta_c$ should be as maximum as possible. As a result:

$$\text{If } \mathbb{1}_{[(k,c) \in \mathcal{J} \cap \mathcal{I}^\star]} = 0 \text{ then } \mathbb{1}_{[k \in \mathcal{K} \cap \mathcal{I}^\star]} = 0 \implies \mu_{k,c} = \|\boldsymbol{\mu}_c\|_\infty = -\eta_c.$$

$\square$

### B.1 SUB-OPTIMALITY GAP

**Proposition B.1** ($S_{\gamma^\star}$ **Property**). *The coefficient $S_{\gamma^\star}$ given in Definition 3 is positive and finite.*

*Proof of Proposition B.1.* The positivity of $S_{\gamma^\star}$ follows immediately from its definition. To establish finiteness, we adopt a sensitivity analysis approach. Consider the mapping:

$$\forall s \in [0, S_{\gamma^\star}], \ s \mapsto y(s) = \max_{\boldsymbol{w} \in \Phi(s)} f(\underline{\boldsymbol{\mu}}, \boldsymbol{w}).$$

It is easy to show that the mapping is decreasing and bounded.

Following same steps as in the proof of Lemma B.1, the Lagrangian function associated to the optimization problem defined by $y(s)$ is given by:

$$\mathcal{L}_s(\boldsymbol{w}, \alpha(s), \beta(s), \eta(s)) = f(\underline{\boldsymbol{\mu}}, \boldsymbol{w}) + \sum_{k \in \mathcal{K}} \alpha_k(s) \Big( g_k(\underline{\boldsymbol{\mu}}, \boldsymbol{w}) - \lambda_k - s \Big)$$

$$+ \sum_{(k,c) \in \mathcal{J}} \beta_{kc}(s) h_k(c, \boldsymbol{w}) + \sum_{c \in \mathcal{C}} \eta_c(s) \Big( q_c(\boldsymbol{w}) - 1 \Big)$$

$$= \sum_{c \in \mathcal{C}} \Big( \underbrace{\boldsymbol{\mu}_c + \Big( \sum_{k=1}^{K} \alpha_k(s) e_{kk} \Big) \boldsymbol{\mu}_c + \sum_{k \in \mathcal{K}} \beta_{kc}(s) e_k + \eta_c(s) \mathbf{1}}_{\boldsymbol{a}_c} \Big)^\top \boldsymbol{w}_c - \sum_{k \in \mathcal{K}} (\lambda_k + s) \alpha_k(s) - \sum_{c \in \mathcal{C}} \eta_c(s)$$

$$= - \sum_{k \in \mathcal{K}} (\lambda_k + s) \alpha_k(s) - \sum_{c \in \mathcal{C}} \eta_c(s)$$

where the last line is due to finiteness of the objective function and the strong duality of the linear programming. Furthermore, using (Enrique Castillo and Castillo, 2006), it follows that:

$$\frac{\partial y}{\partial s} = \sum_{k \in \mathcal{K}} \lambda_k \alpha_k(0).$$

The latter is bounded, given the feasibility of the linear programming $z_0$. This completes the proof. $\square$

**Proposition B.2** ($\mathcal{L}(\mathcal{I})$ **Property**). *For any candidate set $\mathcal{I}$, the coefficient $\mathcal{L}(\mathcal{I})$ given in Definition 5 is positive and finite.*

*Proof of Proposition B.2.* The positivity of $\mathcal{L}(\mathcal{I})$ follows immediately from its definition. To establish finiteness, we adopt a sensitivity analysis approach. Consider the mapping:

$$\forall s \geq s(\mathcal{I}) : s \mapsto z_{\mathcal{I}}(s) = \max_{\underline{w} \in \Psi(s, \mathcal{I})} f(\underline{\mu}, \underline{w}).$$

It is easy to show that the mapping is increasing and bounded.

Following same steps as in the proof of Lemma B.1, the Lagrangian function associated to the optimization problem defined by $z_s$ is given by:

$$\mathcal{L}_s(\underline{w}, \alpha(s), \alpha'(s), \beta(s), \eta(s), \mathcal{I}) = f(\underline{\mu}, \underline{w}) + \sum_{k \in \mathcal{K}} \alpha_k(s) \Big( g_k(\underline{\mu}, \underline{w}) - \lambda_k + s \Big)$$

$$+ \sum_{k \in \mathcal{K} \cap \mathcal{I}} \alpha'_k(s) \Big( - g_k(\underline{\mu}, \underline{w}) + \lambda_k + s \Big) + \sum_{(k,c) \in \mathcal{J} \cap \mathcal{I}} \beta_{kc}(s) h_k(c, \underline{w}) + \sum_{c \in \mathcal{C}} \eta_c(s) \Big( q_c(\underline{w}) - 1 \Big)$$

$$= \sum_{c \in \mathcal{C}} \underbrace{\left( \boldsymbol{\mu}_c + \Big( \sum_{k=1}^{K} (\alpha_k(s) - \alpha'_k(s) \mathbb{1}_{[k \in \mathcal{K} \cap \mathcal{I}]}) e_{kk} \Big) \boldsymbol{\mu}_c + \sum_{k=1}^{K} \beta_{kc}(s) \mathbb{1}_{[(k,c) \in \mathcal{J} \cap \mathcal{I}]} e_k + \eta_c(s) \mathbf{1} \right)^{\top}}_{\boldsymbol{a}_c} \boldsymbol{w}_c$$

$$- \sum_{k \in \mathcal{K}} (\lambda_k - s) \alpha_k(s) + \sum_{k \in \mathcal{K} \cap \mathcal{I}} (\lambda_k + s) \alpha'_k(s) - \sum_{c \in \mathcal{C}} \eta_c$$

$$= - \sum_{k \in \mathcal{K}} (\lambda_k - s) \alpha_k(s) + \sum_{k \in \mathcal{K} \cap \mathcal{I}} (\lambda_k + s) \alpha'_k(s) - \sum_{c \in \mathcal{C}} \eta_c$$

where the last line is due to finiteness of the objective function and the strong duality of the linear programming. Furthermore, using Enrique Castillo and Castillo (2006), it follows that for all $s \geq s(\mathcal{I})$:

$$\frac{\partial z_{\mathcal{I}}}{\partial s} = \sum_{k \in \mathcal{K}} \lambda_k \alpha_k(s(\mathcal{I})) - \sum_{k \in \mathcal{K} \cap \mathcal{I}} \lambda_k \alpha'_k(s(\mathcal{I})).$$

The latter is bounded, given the feasibility of the linear programming $z_{\mathcal{I}}(s(\mathcal{I}))$. This completes the proof. $\square$

**Lemma 3.1** (**Suboptimality Characterization**). *Under Assumption 3,* $\quad \rho(\mathcal{I}^{\star}) = 0 \quad$ *and* $\quad \rho^{\star} > 0$.

*Proof of Lemma 3.1.* The result $\rho(\mathcal{I}^{\star}) = 0$ follows directly from the fact that $\underline{w}^{\star} \in \psi(0, \mathcal{I}^{\star})$ and that $f(\underline{\mu}, \underline{w}^{\star}) = \max_{\underline{w} \in \psi(0, \mathcal{I}^{\star})} f(\underline{\mu}, \underline{w})$.

Let $\mathcal{I} \neq \mathcal{I}^{\star}$, then:

- If $\psi(0, \mathcal{I}) = \emptyset$, then $s(\mathcal{I}) > 0$ and thus $\rho(\mathcal{I}) > 0$.

- Otherwise, if $\psi(0, \mathcal{I}) \neq \emptyset$, then $s(\mathcal{I}) = 0$ and all allocations in $\psi(0, \mathcal{I})$ are feasible for **LP**$(\underline{\mu}, \underline{\mu})$, implying $f(\underline{\mu}, \underline{w}^{\star}) - \max_{\underline{w} \in \psi(0, \mathcal{I})} f(\underline{\mu}, \underline{w}) > 0$, and thus $\rho(\mathcal{I}) > 0$.

Since the number of suboptimal $\mathcal{I}$ is finite (of number $\binom{\kappa + K}{\kappa - |\mathcal{C}|}$ ), it follows that $\rho^{\star} > 0$. $\square$

## C  CONFIDENCE SET CONSTRUCTION

**Proposition 4.1** (**Confidence set**). *Under Assumption 1, let* $\hat{\underline{\mu}}(t)$ *and* $\mathcal{S}_t$ *defined in Eq. 1 and 2, then:*

(i) $\forall t \geq 1, \mathbb{P} \left( \underline{\mu} \in \mathcal{S}_t(\hat{\underline{\mu}}(t), \delta) \right) \geq 1 - \delta$,

(ii) $\forall t \geq 1$, *w.p. at least* $1 - \delta$, *for any* $\underline{w} \in \pi_K^{|\mathcal{C}|}$, $\tilde{\underline{\mu}}(t) \in \mathcal{S}_t(\hat{\underline{\mu}}(t), \delta)$ *and* $k \in \mathcal{K}$,

$$\left| g_k(\tilde{\underline{\mu}}(t), \underline{w}) - g_k(\underline{\mu}, \underline{w}) \right| \leq \rho_k(\underline{\epsilon}(t), \underline{w}), \quad \text{where} \quad \rho_k(\underline{\epsilon}(t), \underline{w}) = \sum_{c \in \mathcal{C}} 2\epsilon_{k,c}(t) w_{k,c}(t),$$

$$\left| f(\tilde{\underline{\mu}}(t), \underline{w}) - f(\underline{\mu}, \underline{w}) \right| \leq \rho(\underline{\epsilon}(t), \underline{w}), \quad\quad\quad \rho(\underline{\epsilon}(t), \underline{w}) = \sum_{k \in [K]} \rho_k(\underline{\epsilon}(t), \underline{w}).$$

*Proof of Proposition 4.1.* The confidence set relies on concentration inequalities to bound the deviation between the empirical mean $\hat{\mu}_{k,c}(t)$ and the true mean $\mu_{k,c}$ for each arm $k$ and context $c$.

Using Hoeffding's inequality, the probability that $\hat{\mu}_{k,c}(t)$ deviates from $\mu_{k,c}$ is bounded as:

$$\Pr\left(|\hat{\mu}_{k,c}(t) - \mu_{k,c}| \le \epsilon_{k,c}(t)\right) \ge 1 - \frac{\delta}{\kappa}.$$

Applying a union bound over all $k$ and $c$, we ensure that:

$$\Pr\left(\forall (k,c), \ |\hat{\mu}_{k,c}(t) - \mu_{k,c}| \le \epsilon_{k,c}(t)\right) \ge 1 - \delta.$$

This establishes that the true parameter $\underline{\mu}$ lies within $\mathcal{S}_t(\hat{\underline{\mu}}(t), \delta)$ with probability at least $1 - \delta$.

Under the consistency property, both $\tilde{\underline{\mu}}(t)$ and $\underline{\mu}$ belong to the confidence set $\mathcal{S}_t(\hat{\underline{\mu}}(t), \delta)$. Therefore, for any $k \in [K]$ and $c \in \mathcal{C}$, we have:

$$\begin{aligned}
|\tilde{\mu}_{k,c}(t) - \mu_{k,c}| &= |\tilde{\mu}_{k,c}(t) - \hat{\mu}_{k,c}(t) + \hat{\mu}_{k,c}(t) - \mu_{k,c}| \\
&\le |\tilde{\mu}_{k,c}(t) - \hat{\mu}_{k,c}(t)| + |\hat{\mu}_{k,c}(t) - \mu_{k,c}| \\
&\le \epsilon_{k,c}(t) + \epsilon_{k,c}(t) \\
&= 2\epsilon_{k,c}(t).
\end{aligned}$$

Expanding $f$ and $g_k$ with respect to their definitions concludes the proof. $\square$

We finish this section by a result on the estimation error that plays a central role in deriving the upper bounds for both **OLP** and **OPLP**.

**Proposition C.1** (Pairwise Estimation Error Upper Bound). *For any arm $k \in \mathcal{K}$ and context $c \in \mathcal{C}$ where at each step $\delta_t = 1/t$, the following bounds hold:*

$$\mathbb{E}\left[\sum_{t=1}^{T} (\epsilon_{k,c}(t) w_{k,c}(t))^2\right] = \mathcal{O}\left(\log(T) \, \mathbb{E}\left[\log(n_{k,c}(T))\right]\right),$$

$$\mathbb{E}\left[\sum_{t=1}^{T} \epsilon_{k,c}(t) w_{k,c}(t)\right] = \mathcal{O}\left(\sqrt{\log(T)} \mathbb{E}\left[\sqrt{n_{k,c}(T)}\right]\right).$$

*Proof of Proposition C.1.* We denote by $\mathcal{F}_t$ the sigma algebra containing the information available at $t$, i.e set $\mathcal{F}_t = \sigma\{\mathcal{H}_{t-1}, c_t\}$. Then:

$$\begin{aligned}
\mathbb{E}\left[\sum_{t=1}^{T} (\epsilon_{k,c}(t) w_{k,c}(t))^2\right] &= \sum_t \mathbb{E}[\epsilon_{k,c}(t)^2 w_{k,c}(t)^2] \\
&\le \sum_t \mathbb{E}\left[\frac{2\log(2\kappa t)}{n_{k,c}(t-1)} w_{k,c}(t)\right] \\
&\le 2\log(2\kappa T) \sum_t \mathbb{E}\left[\mathbb{E}\left[\frac{\mathbb{1}_{\{(k_t, c_t) = (k,c)\}}}{n_{k,c}(t-1)} \Big| \mathcal{F}_{t-1}\right]\right] \\
&\le 2\log(2\kappa T) \mathbb{E}\left[\sum_t \frac{\mathbb{1}_{\{(k_t, c_t) = (k,c)\}}}{n_{k,c}(t-1)}\right] \\
&\le 2\log(2\kappa T) \mathbb{E}\left[\sum_{t;(k_t,c_t)=(k,c)} \frac{1}{n_{k,c}(t-1)}\right] \\
&\le 2\log(2\kappa T) \mathbb{E}\left[\log(n_{k,c}(T) + 1)\right] \qquad \text{(uses Fact } A.2\text{)} \\
&\le \mathcal{O}\left(\log(T) \mathbb{E}\left[\log(n_{k,c}(T))\right]\right)
\end{aligned}$$

$$\mathbb{E}\left[\sum_{t=1}^{T}\epsilon_{k,c}(t)w_{k,c}(t)\right] \leq \sqrt{2\log(2\kappa T)}\mathbb{E}\left[\sum_{t}\frac{w_{k,c}}{\sqrt{n_{k,c}(t-1)}}\right]$$

$$\leq \sqrt{2\log(2\kappa T)}\sum_{t}\mathbb{E}\left[\mathbb{E}\left[\frac{\mathbb{1}_{\{(k_t,c_t)=(k,c)\}}}{\sqrt{n_{k,c}(t-1)}}\bigg|\mathcal{F}_{t-1}\right]\right]$$

$$\leq \sqrt{2\log(2\kappa T)}\mathbb{E}\left[\sum_{t}\frac{\mathbb{1}_{\{(k_t,c_t)=(k,c)\}}}{\sqrt{n_{k,c}(t-1)}}\right]$$

$$\leq \sqrt{\frac{1}{2}\log(2\kappa T)}\mathbb{E}\left[\sqrt{n_{k,c}(T)}+1\right] \qquad \text{(uses Fact } A.1)$$

$$\leq \mathcal{O}\left(\sqrt{\log(T)}\mathbb{E}\left[\sqrt{n_{k,c}(T)}\right]\right)$$

$\square$

## D  PROOF IDEAS FOR THE UPPER-BOUND RESULTS

**Theorem 5.1** (Upper bounds for **OLP**). *Under Assumptions 1, 2 and 3, the performance and constraint regret of **OLP** satisfy:*

$$\mathbb{E}[\mathcal{R}_T] \leq \mathcal{O}\left(\frac{\log(T)^2}{\rho^\star}\right),$$

$$\mathbb{E}[\mathcal{V}_T] \leq \mathcal{O}\left(\frac{\log(T)^2}{\rho^\star} + \sqrt{|\mathcal{K}\cap\mathcal{I}^\star|\log(T)T}\right).$$

**Theorem 5.2** (Upper bounds for **OPLP**). *Under Assumptions 1, 2 and 4, the performance and constraint regret of **OPLP** satisfy:*

$$\mathbb{E}[\mathcal{R}_T] \leq \mathcal{O}\left(\left(\frac{1}{\gamma^{\star 2}} + \frac{1}{\rho^{\star 2}}\right)\log(T)^2 + \sqrt{|\mathcal{K}\cap\mathcal{I}^\star|\log(T)T}\right),$$

$$\mathbb{E}[\mathcal{V}_T] \leq \mathcal{O}\left(\frac{\lambda}{\gamma^{\star 2}}\log(T)^2\right), \text{ where } \lambda = \sum_{k\in\mathcal{K}}\lambda_k.$$

From Section 3, the problem of learning the optimal allocation $\underline{w}^\star$ can be decomposed in two parts: first, identifying the optimal allocation structure - governed by $\mathcal{I}^\star$, and then, determining the optimal weights for such structure. This decomposition allows a finer analysis at the intersection between MAB - finding $\mathcal{I}^\star$, which plays the role of the optimal arm, and linear bandit - finding the optimal structured allocation.

Both **OLP** and **OPLP** recover $\mathcal{I}^\star$ at a logarithmic rate, with convergence speed governed by $\rho^\star$ - which is connected to the notion of gap in MAB: when either algorithm activates an incorrect constraint set ($\mathcal{I}_t \neq \mathcal{I}^\star$), this indicates insufficient estimation precision. Formally, the confidence radius satisfies $\rho(\underline{\epsilon}(t), \underline{w}(t)) \geq \rho^\star$, as shown in Propositions E.1 and E.6.

The optimistic strategy used by **OLP** ensures no performance regret once the optimal constraint set $\mathcal{I}^\star$ is identified, thus leading to logarithmic performance regret overall. When no arms saturate their revenue constraints, this is also the only source of constraint violation. With binding revenue constraints, however, finding the exact structured allocation which satisfies the constraints resembles a linear bandit problem and translates in $\mathcal{O}(\sqrt{T})$ additional term in $\mathcal{V}_T$.

**OPLP** operates in phases and builds on pessimism to propose conservative allocations that satisfies the constraints by design in the second stage (Proposition E.3). The constraint violation $\mathcal{V}_T$ is thus tied to the number of rounds where **LP**($\underline{\mathbf{UCB}}(t), \underline{\mathbf{LCB}}(t)$) is infeasible, which implies insufficient precision as $\rho(\underline{\epsilon}(t), \underline{w}(t)) \geq \gamma^\star$. This occurs only logarithmically often, with the bound scaling inversely with $\gamma^\star$ (Proposition E.2). On the performance side, the regret incurred to identify $\mathcal{I}^\star$ and reach feasibility scales logarithmically (Propositions E.4 andE.6). Once those are met, the pessimistic strategy's surplus allocation to saturating arms in $\mathcal{K}\cap\mathcal{I}^\star$ dominates the regret. The resulting cumulative regret is bounded by: $\mathcal{O}(\sqrt{|\mathcal{K}\cap\mathcal{I}^\star|T})$, as established in Proposition E.5.

# E  Upper-Bounds

## E.1  Results of OLP

**Theorem 5.1** (Upper bounds for **OLP**). *Under Assumptions 1, 2 and 3, the performance and constraint regret of **OLP** satisfy:*

$$\mathbb{E}[\mathcal{R}_T] \leq \mathcal{O}\left(\frac{\log{(T)}^2}{\rho^\star}\right),$$

$$\mathbb{E}[\mathcal{V}_T] \leq \mathcal{O}\left(\frac{\log{(T)}^2}{\rho^\star} + \sqrt{|\mathcal{K} \cap \mathcal{I}^\star| \log{(T)} T}\right).$$

The proof of Theorem 5.1 primarily relies on analyzing the rate at which Algorithm 1 succeeds in identifying the optimal set $\mathcal{I}^\star$, and on understanding the implications of correctly (or incorrectly) identifying this set on both the regret and the constraint violations.

Knowing $\mathcal{I}^\star$ effectively localizes and defines the optimal allocation $\underline{w}^\star$. Hence, having a very tight estimate is strongly tied to activating $\mathcal{I}^\star$ and achieving the optimal allocation. Conversely, activating a suboptimal set $\mathcal{I}_t$ is closely linked to the largeness of the confidence radius $\rho(\underline{\epsilon}(t), \underline{w}(t))$.

**Proposition E.1.** *For all rounds $t$ where **OLP** saturated the wrong set $\mathcal{I}_t \neq \mathcal{I}^\star$, the following inequality holds w.h.p $1 - 1/t$:*

$$\rho(\underline{\epsilon}(t), \underline{w}(t)) \geq \rho^\star.$$

This implies that if a suboptimal $\mathcal{I}_t$ is activated, the corresponding confidence set is necessarily loose, indicating the presence of a non-negligible confidence radius.

*Proof of Proposition E.1.* We rely for the proof on the impact of saturating the wrong set $\mathcal{I}_t \neq \mathcal{I}^\star$ on feasibility and perfomance.

**Infeasibility.** Recall the set $\psi(s, \mathcal{I})$, given in Definition 4:

$$\psi(s, \mathcal{I}) = \left\{ \underline{w} \in \pi_K^{|\mathcal{C}|} \; : \; \begin{array}{l} \forall k \in \mathcal{K}, \quad g_k(\underline{\mu}, \underline{w}) \geq \lambda_k - s, \\ \forall k \in \mathcal{K} \cap \mathcal{I}, \quad g_k(\underline{\mu}, \underline{w}) \leq \lambda_k + s, \\ \forall (k, c) \in \mathcal{J} \cap \mathcal{I}, \quad h_k(c, \underline{w}) = 0 \end{array} \right\}.$$

At round $t$, this set allows to link effectively the chosen allocation $\underline{w}(t)$, the set of activated constraints $\mathcal{I}_t$ and the radius of confidence set.

**Lemma E.1.** *If $\underline{w}(t)$ is the allocation chosen by **OLP** at time $t$, then w.h.p $1 - 1/t$ :*

$$\underline{w}(t) \in \psi(\rho(\underline{\epsilon}(t), \underline{w}(t)), \mathcal{I}_t)$$

*Proof of Lemma E.1.* By the definition of $\underline{\textbf{UCB}}(t)$ and $\underline{w}(t)$, we have:

$$\forall k \in \mathcal{K}, \quad g_k(\underline{\textbf{UCB}}(t), \underline{w}(t)) \geq \lambda_k, \tag{8}$$

$$\forall k \in \mathcal{K} \cap \mathcal{I}_t, \quad g_k(\underline{\textbf{UCB}}(t), \underline{w}(t)) = \lambda_k, \tag{9}$$

$$\underline{w}(t) \in \pi_K^{|\mathcal{C}|}. \tag{10}$$

Given Proposition 4.1, w.h.p $1 - 1/t$, we also have:

$$\forall k \in \mathcal{K}, \quad g_k(\underline{\mu}, \underline{w}(t)) - \rho_k(\underline{\epsilon}(t), \underline{w}(t)) \leq g_k(\underline{\textbf{UCB}}(t), \underline{w}(t))$$
$$\leq g_k(\underline{\mu}, \underline{w}(t)) + \rho_k(\underline{\epsilon}(t), \underline{w}(t)). \tag{11}$$

From (8) and (11), we conclude:

$$\forall k \in \mathcal{K}, \quad g_k(\underline{\mu}, \underline{w}(t)) \geq \lambda_k - \rho_k(\underline{\epsilon}(t), \underline{w}(t)) \geq \lambda_k - \rho(\underline{\epsilon}(t), \underline{w}(t)).$$

Similarly, from (9) and (11):

$$\forall k \in \mathcal{K} \cap \mathcal{I}_t, \quad g_k(\underline{\mu}, \underline{w}(t)) \leq \lambda_k + \rho_k(\underline{\epsilon}(t), \underline{w}(t)).$$

Finally, by the definition of $\mathcal{I}_t$, we also have:

$$\forall (k, c) \in \mathcal{J} \cap \mathcal{I}_t, \quad h_k(c, \underline{w}(t)) = 0,$$

which completes the proof. $\qquad\square$

The inclusion $\underline{\boldsymbol{w}}(t) \in \psi(\rho(\underline{\boldsymbol{\epsilon}}(t), \underline{\boldsymbol{w}}(t)), \mathcal{I}_t)$ established in Lemma E.1 implies that the set $\psi(\rho(\underline{\boldsymbol{\epsilon}}(t), \mathcal{I}_t))$ must contain at least one feasible point. For this to hold, it must be the case that:

$$\rho(\underline{\boldsymbol{\epsilon}}(t), \underline{\boldsymbol{w}}(t)) \geq s(\mathcal{I}_t),$$

where $s(\mathcal{I}_t)$ is given in Definition 4.

**Performance Gap.** Another tension arises due to performance. We consider the following problem:

$$z_s(\underline{\boldsymbol{\mu}}, \mathcal{I}) = \max_{\underline{\boldsymbol{w}} \in \psi(s, \mathcal{I})} f(\underline{\boldsymbol{\mu}}, \underline{\boldsymbol{w}}).$$

Given Definition 5, we have:

$$z_{\rho(\underline{\boldsymbol{\epsilon}}(t), \underline{\boldsymbol{w}}(t))}(\underline{\boldsymbol{\mu}}, \mathcal{I}_t) - z_{s(\mathcal{I}_t)}(\underline{\boldsymbol{\mu}}, \mathcal{I}_t) \leq \mathcal{L}(\mathcal{I}_t) \left( \rho(\underline{\boldsymbol{\epsilon}}(t), \underline{\boldsymbol{w}}(t)) - s(\mathcal{I}_t) \right).$$

On the other hand, it holds that:

$$f(\underline{\boldsymbol{\mu}}, \underline{\boldsymbol{w}}(t)) \leq z_{\rho(\underline{\boldsymbol{\epsilon}}(t), \underline{\boldsymbol{w}}(t))}(\underline{\boldsymbol{\mu}}, \mathcal{I}_t),$$

and by optimism:

$$f(\underline{\boldsymbol{\mu}}, \underline{\boldsymbol{w}}^\star) - \rho(\underline{\boldsymbol{\epsilon}}(t), \underline{\boldsymbol{w}}(t)) \leq f(\underline{\boldsymbol{\mu}}, \underline{\boldsymbol{w}}(t)).$$

Hence, we obtain:

$$f(\underline{\boldsymbol{\mu}}, \underline{\boldsymbol{w}}^\star) - \rho(\underline{\boldsymbol{\epsilon}}(t), \underline{\boldsymbol{w}}(t)) - z_{s(\mathcal{I}_t)}(\underline{\boldsymbol{\mu}}, \mathcal{I}_t) \leq \mathcal{L}(\mathcal{I}_t) \left( \rho(\underline{\boldsymbol{\epsilon}}(t), \underline{\boldsymbol{w}}(t)) - s(\mathcal{I}_t) \right).$$

This implies:

$$\frac{f(\underline{\boldsymbol{\mu}}, \underline{\boldsymbol{w}}^\star) - z_{s(\mathcal{I}_t)}(\underline{\boldsymbol{\mu}}, \mathcal{I}_t) + \mathcal{L}(\mathcal{I}_t) s(\mathcal{I}_t)}{1 + \mathcal{L}(\mathcal{I}_t)} \leq \rho(\underline{\boldsymbol{\epsilon}}(t), \underline{\boldsymbol{w}}(t))$$

$$\implies \frac{f(\underline{\boldsymbol{\mu}}, \underline{\boldsymbol{w}}^\star) - z_{s(\mathcal{I}_t)}(\underline{\boldsymbol{\mu}}, \mathcal{I}_t)}{\max(1, S_{\gamma^\star}) + \mathcal{L}(\mathcal{I}_t)} \leq \rho(\underline{\boldsymbol{\epsilon}}(t), \underline{\boldsymbol{w}}(t))$$

$$\implies \rho(\mathcal{I}_t) \leq \rho(\underline{\boldsymbol{\epsilon}}(t), \underline{\boldsymbol{w}}(t))$$

Hence, if $\mathcal{I}_t$ is sub-optimal then $\rho(\mathcal{I}_t) \geq \rho^\star$ which completes the proof.

$\square$

### E.1.1 REGRET OF **OLP**

To prove the upper bound on the regret of **OLP**, we examine the per-round regret incurred when the algorithm activates (or fails to activate) the optimal set of constraints, $\mathcal{I}^\star$. Note that for any round $t$ of **OLP**, it holds :

$$\text{w.h.p } 1 - {}^1\!/_t, \quad f(\underline{\boldsymbol{\mu}}, \underline{\boldsymbol{w}}^\star) - f(\underline{\boldsymbol{\mu}}, \underline{\boldsymbol{w}}(t)) \leq \rho(\underline{\boldsymbol{\epsilon}}(t), \underline{\boldsymbol{w}}(t)) \tag{12}$$

*Proof of Equation 12.* Optimism ensures that $f(\underline{\boldsymbol{\mu}}, \underline{\boldsymbol{w}}^\star) \leq f(\mathbf{UCB}(t), \underline{\boldsymbol{w}}(t))$. By Proposition 4.1, we have $f(\mathbf{UCB}(t), \underline{\boldsymbol{w}}(t)) \leq f(\underline{\boldsymbol{\mu}}, \underline{\boldsymbol{w}}(t)) + \rho(\underline{\boldsymbol{\epsilon}}(t), \underline{\boldsymbol{w}}(t))$. Combining these two steps concludes the proof. $\square$

**For Suboptimal $\mathcal{I}_t \neq \mathcal{I}^\star$:** Recall the Proposition E.1 on $\rho^\star$, then w.h.p $1 - {}^1\!/_t$:

$$f(\underline{\boldsymbol{\mu}}, \underline{\boldsymbol{w}}^\star) - f(\underline{\boldsymbol{\mu}}, \underline{\boldsymbol{w}}(t)) \stackrel{Eq\ (12)}{\leq} \rho(\underline{\boldsymbol{\epsilon}}(t), \underline{\boldsymbol{w}}(t)) = \rho(\underline{\boldsymbol{\epsilon}}(t), \underline{\boldsymbol{w}}(t)) \mathbb{1}_{[\rho(\underline{\boldsymbol{\epsilon}}(t), \underline{\boldsymbol{w}}(t)) \geq \rho^\star]} \leq \frac{\rho(\underline{\boldsymbol{\epsilon}}(t), \underline{\boldsymbol{w}}(t))^2}{\rho^\star} \tag{13}$$

**For optimal $\mathcal{I}_t = \mathcal{I}^\star$:** Both $\underline{w}(t)$ and $\underline{w}^\star$ share the same localization of non-zero entries. However, the estimate $\underline{\mu}(t)$ used at time $t$ may lead to differences in their values. Given that $\mathcal{K} \cap \mathcal{I}_t = \mathcal{K} \cap \mathcal{I}^\star$:

$$\forall k \in \mathcal{I}_t \cap \mathcal{K}, \quad g_k(\underline{\mathbf{UCB}}(t), \underline{w}(t)) = \lambda_k = g_k(\underline{\mu}, \underline{w}^\star)$$
$$\implies \forall k \in \mathcal{I}_t \cap \mathcal{K}, \quad g_k(\underline{\mu}, \underline{w}(t)) \leq g_k(\underline{\mu}, \underline{w}^\star) \tag{14}$$

Notice that:

$$f(\underline{\mu}, \underline{w}^\star) = \sum_{c \in \mathcal{C}} \sum_{k=1}^{K} \mu_{k,c} w_{k,c}$$

$$\stackrel{(a)}{=} \sum_{c \in \mathcal{C}} \left( w_{k_c^\star, c} \|\boldsymbol{\mu}_c\|_\infty + \sum_{k \neq k_c^\star} \mu_{k,c} w_{k,c} \right)$$

$$= \sum_{c \in \mathcal{C}} \left( \|\boldsymbol{\mu}_c\|_\infty - \sum_{k \neq k_c^\star} \Delta_{k,c} w_{k,c} \right)$$

$$\stackrel{(b)}{=} \sum_{c \in \mathcal{C}} \left( \|\boldsymbol{\mu}_c\|_\infty - \sum_{k \in \mathcal{K} \cap \mathcal{I}^\star} \Delta_{k,c} w_{k,c} \right)$$

where in (a), $k_c^\star = \mathrm{argmax}_{k \in [K]} \mu_{k,c}$, and (b) uses Lemma B.1, which shows that in a given context, the only non-saturating arm that may have non-zero probability is the best arm in that context.

And given that $\mathcal{I}_t = \mathcal{I}^\star$, then similarly:

$$f(\underline{\mu}, \underline{w}(t)) = \sum_{c \in \mathcal{C}} \left( \|\boldsymbol{\mu}_c\|_\infty - \sum_{k \in \mathcal{K} \cap \mathcal{I}^\star} \Delta_{k,c} w_{k,c}(t) \right)$$

Hence:

$$f(\underline{\mu}, \underline{w}^\star) - f(\underline{\mu}, \underline{w}(t)) = \sum_{c \in \mathcal{C}} \sum_{k \in \mathcal{K} \cap \mathcal{I}^\star} \Delta_{k,c} \left( w_{k,c}(t) - w_{k,c} \right) \leq 0 \tag{15}$$

This demonstrates that, the UCB-based approach ensures nor regret between the optimal solution $\underline{w}^\star$ and the estimated solution $\underline{w}(t)$, if $\mathcal{I}_t = \mathcal{I}^\star$.

Thus, the regret is upper bounded by:

$$\mathcal{R}_T \leq \sum_{t=1}^{T} \left( f(\underline{\mu}, \underline{w}^\star) - f(\underline{\mu}, \underline{w}(t)) \right)_+$$

$$\leq \sum_{t=1}^{T} \left( f(\underline{\mu}, \underline{w}^\star) - f(\underline{\mu}, \underline{w}(t)) \right)_+ \mathbb{1}_{[\mathcal{I}_t = \mathcal{I}^\star]} + \sum_{t=1}^{T} \left( f(\underline{\mu}, \underline{w}^\star) - f(\underline{\mu}, \underline{w}(t)) \right)_+ \mathbb{1}_{[\mathcal{I}_t \neq \mathcal{I}^\star]}$$

$$\stackrel{Eq (15)}{\leq} \sum_{t=1}^{T} \left( f(\underline{\mu}, \underline{w}^\star) - f(\underline{\mu}, \underline{w}(t)) \right)_+ \mathbb{1}_{[\mathcal{I}_t \neq \mathcal{I}^\star]}$$

$$\implies \mathbb{E}\left[\mathcal{R}_T\right] \leq \mathbb{E}\left[ \sum_{t=1}^{T} \left( f(\underline{\mu}, \underline{w}^\star) - f(\underline{\mu}, \underline{w}(t)) \right)_+ \mathbb{1}_{[\mathcal{I}_t \neq \mathcal{I}^\star]} \right]$$

For each round, we decompose the round wise regret by analyzing two distinct scenarios: the good event (denoted by $\mathcal{GE}$) occurring with high probability $1 - 1/t$ as guaranteed by Proposition 4.1, and the bad event occurring with complementary probability $1/t$. In the latter case, we conservatively bounded the roundwise regret by the quantity $\mu = \sum_{c \in \mathcal{C}} \|\boldsymbol{\mu}_c\|_\infty$, which provides a worst-case losses.

Hence:

$$\mathbb{E}\left[\mathcal{R}_T\right] \leq \mathbb{E}\left[\sum_{t=1}^{T}\left(f(\boldsymbol{\mu},\boldsymbol{w}^\star)-f(\boldsymbol{\mu},\boldsymbol{w}(t))\right)_+\mathbb{1}_{[\mathcal{I}_t\neq\mathcal{I}^\star]}\right]$$

$$\leq \sum_{t=1}^{T}\mathbb{E}\left[\left(f(\boldsymbol{\mu},\boldsymbol{w}^\star)-f(\boldsymbol{\mu},\boldsymbol{w}(t))\right)_+\mathbb{1}_{[\mathcal{I}_t\neq\mathcal{I}^\star]}\mid\mathcal{GE}\right].1+\mathbb{E}\left[\left(f(\boldsymbol{\mu},\boldsymbol{w}^\star)-f(\boldsymbol{\mu},\boldsymbol{w}(t))\right)_+\mathbb{1}_{[\mathcal{I}_t\neq\mathcal{I}^\star]}\mid\overline{\mathcal{GE}}\right]1/t$$

$$\overset{(a)}{\leq} \sum_{t=1}^{T}\mathbb{E}\left[\frac{\rho(\boldsymbol{\epsilon}(t),\boldsymbol{w}(t))^2}{\rho^\star}\right]+\frac{\mu}{t}$$

$$\leq \sum_{t=1}^{T}\mathbb{E}\left[\frac{\rho(\boldsymbol{\epsilon}(t),\boldsymbol{w}(t))^2}{\rho^\star}\right]+\mathcal{O}\left(\mu\log(T)\right)$$

Where in (a), we use Equation (13) for the first term and as discussed we upper bound the roundwise regret by $\mu$ for the second term. Hence, to control the expected regret, it remains to control

$$\frac{1}{\rho^\star}\mathbb{E}\left[\sum_{t=1}^{T}\rho(\boldsymbol{\epsilon}(t),\boldsymbol{w}(t))^2\right]=\frac{1}{\rho^\star}\mathbb{E}\left[\sum_{t=1}^{T}\left(\sum_{k,c}\epsilon_{k,c}(t)w_{k,c}(t)\right)^2\right]$$

$$\leq \frac{\kappa}{\rho^\star}\sum_{k,c}\mathbb{E}\left[\sum_{t=1}^{T}\left(\epsilon_{k,c}(t)w_{k,c}(t)\right)^2\right]$$

$$\overset{\textbf{Prop.}C.1}{\leq} \mathcal{O}\left(\frac{\kappa}{\rho^\star}\log(T)\sum_{k,c}\log\left(n_{k,c}(T)\right)\right)$$

$$\overset{\log(.)\text{concavity}}{\leq} \mathcal{O}\left(\frac{\kappa^2}{\rho^\star}\log(T)^2\right) \tag{16}$$

which concludes the proof.

### E.1.2 Constraints Violation of **OLP**

Using Proposition 4.1, w.h.p $1-1/t$, for any arm $k$, the constraints evaluated using the estimated and true means satisfy the following relationship:

$$g_k(\underline{\textbf{UCB}}(t),\boldsymbol{w}(t))\leq g_k(\boldsymbol{\mu},\boldsymbol{w}(t))+\rho_k(\boldsymbol{\epsilon}(t),\boldsymbol{w}(t)).$$

Additionally, by the feasibility condition of the solution to $\textbf{LP}(\underline{\textbf{UCB}}(t),\underline{\textbf{UCB}}(t))$, we have:

$$g_k(\underline{\textbf{UCB}}(t),\boldsymbol{w}(t))\geq\lambda_k,\quad\forall k\in\mathcal{K}.$$

Combining the above results yields:

$$\lambda_k-g_k(\boldsymbol{\mu},\boldsymbol{w}(t))\leq\rho_k(\boldsymbol{\epsilon}(t),\boldsymbol{w}(t)),\quad\forall k\in\mathcal{K}. \tag{17}$$

Now, consider rounds $t$ such that $\mathcal{I}_t=\mathcal{I}^\star$:

1. If $k\in\mathcal{K}\cap\mathcal{I}^\star$, then $g_k(\boldsymbol{\mu},\boldsymbol{w}^\star)=\lambda_k$. On the other hand, given (14), we have $g_k(\boldsymbol{\mu},\boldsymbol{w}^\star)\geq g_k(\boldsymbol{\mu},\boldsymbol{w}(t))$. Thus,
$$\lambda_k\geq g_k(\boldsymbol{\mu},\boldsymbol{w}(t)).$$

2. If $k\notin\mathcal{K}\cap\mathcal{I}^\star$, then $\lambda_k\leq g_k(\boldsymbol{\mu},\boldsymbol{w}^\star)$. Furthermore, we have
$$g_k(\boldsymbol{\mu},\boldsymbol{w}^\star)\leq g_k(\boldsymbol{\mu},\boldsymbol{w}(t))\implies\lambda_k\leq g_k(\boldsymbol{\mu},\boldsymbol{w}(t)).$$

Consequently, when $\mathcal{I}_t=\mathcal{I}^\star$, the violation arises only from saturating arms and is given by:

$$\mathcal{V}(t)=\sum_{k\in\mathcal{K}\cap\mathcal{I}^\star}\lambda_k-g_k(\boldsymbol{\mu},\boldsymbol{w}(t))\overset{(17)}{\leq}\sum_{k\in\mathcal{K}\cap\mathcal{I}^\star}\rho_k(\boldsymbol{\epsilon}(t),\boldsymbol{w}(t)).$$



Thus, the total constraint violation up to time $T$ can be expressed as:

$$
\begin{aligned}
\mathcal{V}_T &= \sum_{t=1}^{T} \sum_{k \in \mathcal{K}} \left( \lambda_k - g_k(\underline{\mu}, \underline{w}(t)) \right)_+ \\
&= \sum_{t=1}^{T} \mathbb{1}_{[\mathcal{I}_t = \mathcal{I}^\star]} \sum_{k \in \mathcal{K}} \left( \lambda_k - g_k(\underline{\mu}, \underline{w}(t)) \right)_+ + \sum_{t=1}^{T} \mathbb{1}_{[\mathcal{I}_t \neq \mathcal{I}^\star]} \sum_{k \in \mathcal{K}} \left( \lambda_k - g_k(\underline{\mu}, \underline{w}(t)) \right)_+ \\
&= \underbrace{\sum_{t=1}^{T} \mathbb{1}_{[\mathcal{I}_t = \mathcal{I}^\star]} \sum_{k \in \mathcal{K} \cap \mathcal{I}^\star} \left( \lambda_k - g_k(\underline{\mu}, \underline{w}(t)) \right)_+}_{\mathcal{A}} + \underbrace{\sum_{t=1}^{T} \mathbb{1}_{[\mathcal{I}_t \neq \mathcal{I}^\star]} \sum_{k \in \mathcal{K}} \left( \lambda_k - g_k(\underline{\mu}, \underline{w}(t)) \right)_+}_{\mathcal{B}}
\end{aligned}
$$

We proceed by establishing upper bounds for the expected values of both $\mathcal{A}$ and $\mathcal{B}$. Our analysis decomposes these quantities under two scenarios: the good event $\mathcal{GE}$ from Proposition 4.1 and its complementary event. Let $\lambda = \sum_{K \in \mathcal{K}} \lambda_k$ denote the worst-case constraint violation that may occur in any given round $t$.

$$
\begin{aligned}
\mathbb{E}[\mathcal{A}] &= \mathbb{E} \left[ \sum_{t=1}^{T} \mathbb{1}_{[\mathcal{I}_t = \mathcal{I}^\star]} \sum_{k \in \mathcal{K} \cap \mathcal{I}^\star} \left( \lambda_k - g_k(\underline{\mu}, \underline{w}(t)) \right)_+ \right] \\
&\leq \sum_{t=1}^{T} \mathbb{E} \left[ \mathbb{1}_{[\mathcal{I}_t = \mathcal{I}^\star]} \sum_{k \in \mathcal{K} \cap \mathcal{I}^\star} \left( \lambda_k - g_k(\underline{\mu}, \underline{w}(t)) \right)_+ \mid \mathcal{GE} \right] .1 + \mathbb{E} \left[ \mathbb{1}_{[\mathcal{I}_t = \mathcal{I}^\star]} \sum_{k \in \mathcal{K} \cap \mathcal{I}^\star} \left( \lambda_k - g_k(\underline{\mu}, \underline{w}(t)) \right)_+ \mid \overline{\mathcal{GE}} \right] 1/t \\
&\leq \sum_{t=1}^{T} \mathbb{E} \left[ \mathbb{1}_{[\mathcal{I}_t = \mathcal{I}^\star]} \sum_{k \in \mathcal{K} \cap \mathcal{I}^\star} \rho_k(\underline{\epsilon}(t), \underline{w}(t)) \right] + \frac{\lambda}{t} \\
&\leq \sum_{k \in \mathcal{K} \cap \mathcal{I}^\star} \mathbb{E} \left[ \sum_{t=1}^{T} \rho_k(\underline{\epsilon}(t), \underline{w}(t)) \right] + \mathcal{O}\left( \lambda \log(T) \right) \\
&\leq \sum_{k \in \mathcal{K} \cap \mathcal{I}^\star} \sum_{c \in \mathcal{C}} \mathbb{E} \left[ \sum_{t=1}^{T} \epsilon_{k,c}(t) w_{k,c}(t) \right] + \mathcal{O}\left( \lambda \log(T) \right) \\
&\overset{Prop\ C.1}{\leq} \mathcal{O}\left( \mathbb{E} \left[ \log(T) \sum_{k \in \mathcal{K} \cap \mathcal{I}^\star} \sum_{c \in \mathcal{C}} \sqrt{n_{k,c}(T)} \right] \right) + \mathcal{O}\left( \lambda \log(T) \right) \\
&\overset{\sqrt{.}\ concavity}{\leq} \mathcal{O}\left( \sqrt{|\mathcal{C}| . |\mathcal{K} \cap \mathcal{I}^\star| \log(T) T} + \lambda \log(T) \right)
\end{aligned}
\tag{18}
$$

$$
\begin{aligned}
\mathbb{E}[\mathcal{B}] &= \mathbb{E} \left[ \sum_{t=1}^{T} \mathbb{1}_{[\mathcal{I}_t \neq \mathcal{I}^\star]} \sum_{k \in \mathcal{K}} \left( \lambda_k - g_k(\underline{\mu}, \underline{w}(t)) \right)_+ \right] \\
&\leq \sum_{t=1}^{T} \mathbb{E} \left[ \mathbb{1}_{[\mathcal{I}_t \neq \mathcal{I}^\star]} \rho(\underline{\epsilon}(t), \underline{w}(t)) \mid \mathcal{GE} \right] .1 + \lambda \mathbb{E} \left[ \mathbb{1}_{[\mathcal{I}_t \neq \mathcal{I}^\star]} \mid \overline{\mathcal{GE}} \right] 1/t \\
&\leq \sum_{t=1}^{T} \mathbb{E} \left[ \rho(\underline{\epsilon}(t), \underline{w}(t)) \mathbb{1}_{[\rho(\underline{\epsilon}(t), \underline{w}(t)) \geq \rho^\star]} \mid \mathcal{GE} \right] + \lambda/t \\
&\leq \sum_{t=1}^{T} \mathbb{E} \left[ \frac{\rho(\underline{\epsilon}(t), \underline{w}(t))^2}{\rho^\star} \right] + \mathcal{O}\left( \lambda \log(T) \right) \\
&\overset{same\ as\ Eq\ (16)}{\leq} \mathcal{O}\left( \frac{\kappa^2}{\rho^\star} \log(T)^2 \right)
\end{aligned}
$$

Combining $\mathbb{E}[\mathcal{A}] + \mathbb{E}[\mathcal{B}]$ concludes the proof.

## E.2 Results of OPLP

**Theorem 5.2** (Upper bounds for **OPLP**). *Under Assumptions 1, 2 and 4, the performance and constraint regret of **OPLP** satisfy:*

$$\mathbb{E}[\mathcal{R}_T] \leq \mathcal{O}\left(\left(\frac{1}{\gamma^{\star 2}} + \frac{1}{\rho^{\star 2}}\right) \log(T)^2 + \sqrt{|\mathcal{K} \cap \mathcal{I}^\star| \log(T) T}\right),$$

$$\mathbb{E}[\mathcal{V}_T] \leq \mathcal{O}\left(\frac{\lambda}{\gamma^{\star 2}} \log(T)^2\right), \text{ where } \lambda = \sum_{k \in \mathcal{K}} \lambda_k.$$

The specificity of **OPLP** lies in its use of pessimistic estimates as parameters for the constraints, introducing a safety margin that enhances constraint satisfaction. However, step 6 is not guaranteed to be feasible from the outset. For this reason, the algorithm relies on an optimistic approach as a fallback. Under Assumption 4, one can control the number of rounds during which the pessimistic step is infeasible. Recall the Definition 2 of $\gamma^\star$, that quantifies the strong feasibility of the problem. It is crucial to control the number of rounds pessimism is infeasible.

**Proposition E.2.** *Consider the event $\mathcal{E}(t) = \{\rho(\underline{\epsilon}(t), \underline{w}(t)) \geq \gamma\}$, and define $\mathcal{E}_T = \sum_{t=1}^{T} \mathbb{1}_{[\mathcal{E}(t)]}$. Let $\tau$ denote the number of rounds in which step 9 of Algorithm 2 is executed. Then, under **OPLP**, the following holds:*

$$\tau \leq \mathcal{E}_T,$$

$$\mathbb{E}[\mathcal{E}_T] \leq \mathcal{O}\left(\frac{2\kappa^2}{\gamma^2} \log(T) \log\left(\frac{2\kappa}{\delta}\right)\right).$$

*Proof of Proposition E.2.* Consider the event $\mathcal{E}(t) = \{\rho(\underline{\epsilon}(t), \underline{w}(t)) \geq \gamma\}$

To ensure the feasibility of the **LCB**, the following suffices to hold:

$$\exists \underline{w} \in \pi_K^{|\mathcal{C}|}, \quad \forall k \in [K], \quad g_k(\underline{\mu}, \underline{w}) - \rho_k(\underline{\epsilon}, \underline{w}) \geq \lambda_k.$$

Thus, it suffices to ensure that:

$$\forall k \in \mathcal{K}, \quad \rho_k(\underline{\epsilon}, \underline{w}) \leq \gamma,$$

Implying that:

$$\tau \leq \sum_{t=1}^{T} \mathbb{1}_{[\mathcal{E}(t)]} = \mathcal{E}_T \leq \sum_{t=1}^{T} \mathbb{1}_{[\rho(\underline{\epsilon}(t), \underline{w}(t)) \geq \gamma]} \leq \sum_{t=1}^{T} \frac{\rho(\underline{\epsilon}(t), \underline{w}(t))^2}{\gamma^2}$$

$$\implies \mathbb{E}[\tau] \leq \mathbb{E}[\mathcal{E}_T] \leq \frac{1}{\gamma^2} \sum_{t=1}^{T} \mathbb{E}\left[\rho(\underline{\epsilon}(t), \underline{w}(t))^2\right] \overset{\text{same as Eq (16)}}{\leq} \mathcal{O}\left(\frac{\kappa^2}{\gamma^2} \log(T)^2\right)$$

$\square$

### E.2.1 Constraints Violation of OPLP

The use of pessimistic estimates for the constraints is advantageous in terms of limiting constraint violations.

**Proposition E.3.** *If at round $t$, the problem $LP(\underline{UCB}(t), \underline{LCB}(t))$ is feasible, then w.h.p $1 - 1/t$, the corresponding constraint violation is zero.*

*Proof of Proposition E.3.* Suppose that step 6 of **OPLP** is feasible at round $t$, and let $\underline{w}(t)$ denote the corresponding solution. Then, by feasibility, we have:

$$\forall k \in \mathcal{K}, \quad g_k(\underline{LCB}(t), \underline{w}(t)) \geq \lambda_k.$$

By the pessimism property of the lower confidence bounds in Proposition 4.1, we know that w.h.p $1 - 1/t$:

$$g_k(\underline{\mu}, \underline{w}(t)) \geq g_k(\underline{LCB}(t), \underline{w}(t)) \geq \lambda_k.$$

Hence, the constraints are satisfied under the true means $\underline{\mu}$, implying that the constraint violation is zero. $\square$

Thus, at each round $t$:

- If step 6 is feasible, then the per-round constraint violation is zero.
- Otherwise, the per-round constraint violation is at most $\lambda = \sum_{k \in \mathcal{K}} \lambda_k$.

Using Proposition E.2, we have:

$$\mathcal{V}_T \leq \sum_{t=1}^{T} (\lambda_k - g_k(\underline{\boldsymbol{\mu}}, \underline{\boldsymbol{w}}^\star))_+ \left( \mathbb{1}_{[\mathcal{E}(t)]} + \mathbb{1}_{[\overline{\mathcal{E}(t)}]} \mathbb{1}_{[\mathcal{GE}]} + \mathbb{1}_{[\overline{\mathcal{E}(t)}]} \mathbb{1}_{[\overline{\mathcal{GE}}]} \right),$$

$$\implies \mathbb{E}[\mathcal{V}_T] \leq \underbrace{\lambda \mathbb{E}[\mathcal{E}_T]}_{(a)} + \underbrace{0}_{(b)} + \underbrace{\sum_{t=1}^{T} \frac{\lambda}{t}}_{(c)} \leq \mathcal{O}\left( \frac{\kappa^2 \lambda}{\gamma^2} \log{(T)}^2 \right).$$

Where (a) is the consequence of Proposition E.2, (b) is the consequence of Proposition E.3 and (c) is the result of the low probability event of Proposition 4.1. This concludes the proof of the upper bound on the cumulative constraints violation under **OPLP**.

### E.2.2 REGRET OF **OPLP**

The regret of **OPLPs** can be decomposed based on whether step 6 is feasible or not. Once the pessimistic step is feasible, a further decomposition considers whether the optimal set of constraints, $\mathcal{I}^\star$, is saturated or not.

$$\mathcal{R}_T = \sum_{t=1}^{T} \left( f(\underline{\boldsymbol{\mu}}, \underline{\boldsymbol{w}}^\star) - f(\underline{\boldsymbol{\mu}}, \underline{\boldsymbol{w}}(t)) \right)_+$$

$$= \sum_{t=1}^{T} \left( f(\underline{\boldsymbol{\mu}}, \underline{\boldsymbol{w}}^\star) - f(\underline{\boldsymbol{\mu}}, \underline{\boldsymbol{w}}(t)) \right)_+ \mathbb{1}_{[\mathcal{E}(t)]} + \sum_{t=1}^{T} \left( f(\underline{\boldsymbol{\mu}}, \underline{\boldsymbol{w}}^\star) - f(\underline{\boldsymbol{\mu}}, \underline{\boldsymbol{w}}(t)) \right)_+ \mathbb{1}_{[\overline{\mathcal{E}(t)}]}$$

$$= \underbrace{\sum_{t=1}^{T} \left( f(\underline{\boldsymbol{\mu}}, \underline{\boldsymbol{w}}^\star) - f(\underline{\boldsymbol{\mu}}, \underline{\boldsymbol{w}}(t)) \right)_+ \mathbb{1}_{[\mathcal{E}(t)]}}_{\mathcal{A}_1} + \underbrace{\sum_{t=1}^{T} \left( f(\underline{\boldsymbol{\mu}}, \underline{\boldsymbol{w}}^\star) - f(\underline{\boldsymbol{\mu}}, \underline{\boldsymbol{w}}(t)) \right)_+ \mathbb{1}_{[\overline{\mathcal{E}(t)}]} \mathbb{1}_{[\mathcal{I}_t = \mathcal{I}^\star]}}_{\mathcal{A}_2}$$

$$+ \underbrace{\sum_{t=1}^{T} \left( f(\underline{\boldsymbol{\mu}}, \underline{\boldsymbol{w}}^\star) - f(\underline{\boldsymbol{\mu}}, \underline{\boldsymbol{w}}(t)) \right)_+ \mathbb{1}_{[\overline{\mathcal{E}(t)}]} \mathbb{1}_{[\mathcal{I}_t \neq \mathcal{I}^\star]}}_{\mathcal{A}_3}$$

Then we proceed by upper-bounding each term $\mathcal{A}_1, \mathcal{A}_2$ and $\mathcal{A}_3$.

**Upper-bounding $\mathcal{A}_1$.** This quantifies the regret induced during the infeasibility of the pessimistic approach.

**Proposition E.4.** *Under **OPLP**, we have:*

$$\mathbb{E}[\mathcal{A}_1] \leq \mathcal{O}\left( \frac{\kappa^2}{\gamma^2} \log{(T)}^2 \right).$$

*Proof of Proposition E.4.* The per round regret is upperbounded by $\mu = \sum_{c \in \mathcal{C}} \|\boldsymbol{\mu}_c\|_\infty$. Hence:

$$\mathbb{E}[\mathcal{A}_1] = \mathbb{E}\left[ \sum_{t=1}^{T} \left( f(\underline{\boldsymbol{\mu}}, \underline{\boldsymbol{w}}^\star) - f(\underline{\boldsymbol{\mu}}, \underline{\boldsymbol{w}}(t)) \right)_+ \mathbb{1}_{[\mathcal{E}(t)]} \right]$$

$$\leq \mathbb{E}\left[ \sum_{t=1}^{T} \mu \mathbb{1}_{[\mathcal{E}(t)]} \right]$$

$$\overset{(a)}{\leq} \mathcal{O}\left( \frac{\kappa^2}{\gamma^2} \log{(T)}^2 \right)$$

where (a) is based on Proposition E.2. $\qquad\square$

**Upper-bounding $\mathcal{A}_2$.** This term corresponds to the rounds where step 6 of **OPLP** is feasible and the optimal constraints are saturated, i.e., $\mathcal{I}_t = \mathcal{I}^\star$. During these rounds, the algorithm safely activates the saturating arms, i.e., $k \in \mathcal{K} \cap \mathcal{I}^\star$, by allocating them more budget, which induces the regret.

**Proposition E.5.** *Under **OPLP**, we have:*

$$\mathbb{E}\left[\mathcal{A}_2\right] \leq \mathcal{O}\left(\sqrt{|\mathcal{C}|\,|\mathcal{K} \cap \mathcal{I}^\star|\log\left(T\right)T}\right).$$

*Proof of Proposition E.5.* For the second phase, when using **LCB** becomes possible, we consider rounds $t$ where $\mathcal{I}_t = \mathcal{I}^\star$. This implies that both $\mathbf{LP}(\underline{\mu}, \underline{\mu})$ and $\mathbf{LP}(\underline{\mu}, \underline{\mathbf{LCB}}(t))$ saturate the same arms, i.e:

$$\forall k \in \mathcal{I}_t \cap \mathcal{K}, \quad g_k(\underline{\mathbf{LCB}}(t), \underline{\mathbf{w}}(t)) = \lambda_k = g_k(\underline{\mu}, \underline{\mathbf{w}}^\star)$$

$$\implies \text{w.h.p: } 1 - 1/t, \forall k \in \mathcal{I}_t \cap \mathcal{K}, \quad g_k(\underline{\mu}, \underline{\mathbf{w}}(t)) - \rho_k(\underline{\epsilon}(t), \underline{\mathbf{w}}(t)) \leq g_k(\underline{\mu}, \underline{\mathbf{w}}^\star)$$

$$\implies \text{w.h.p: } 1 - 1/t, \forall k \in \mathcal{I}_t \cap \mathcal{K}, \quad g_k(\underline{\mu}, \underline{\mathbf{w}}(t)) - g_k(\underline{\mu}, \underline{\mathbf{w}}^\star) \leq \rho_k(\underline{\epsilon}(t), \underline{\mathbf{w}}(t)) \qquad (19)$$

Notice that:

$$f(\underline{\mu}, \underline{\mathbf{w}}^\star) = \sum_{c \in \mathcal{C}} \sum_{k=1}^{K} \mu_{k,c} w_{k,c}$$

$$\overset{(a)}{=} \sum_{c \in \mathcal{C}} \left( w_{k_c^\star, c} \|\boldsymbol{\mu}_c\|_\infty + \sum_{k \neq k_c^\star} \mu_{k,c} w_{k,c} \right)$$

$$= \sum_{c \in \mathcal{C}} \left( \|\boldsymbol{\mu}_c\|_\infty - \sum_{k \neq k_c^\star} \Delta_{k,c} w_{k,c} \right)$$

$$\overset{(b)}{=} \sum_{c \in \mathcal{C}} \left( \|\boldsymbol{\mu}_c\|_\infty - \sum_{k \in \mathcal{K} \cap \mathcal{I}^\star} \Delta_{k,c} w_{k,c} \right)$$

where in (a), $k_c^\star = \operatorname{argmax}_{k \in [K]} \mu_{k,c}$, and (b) uses Lemma B.1, which shows that in a given context, the only non-saturating arm that may have non-zero probability is the best arm in that context.

And given that $\mathcal{I}_t = \mathcal{I}^\star$, then similarly:

$$f(\underline{\mu}, \underline{\mathbf{w}}(t)) = \sum_{c \in \mathcal{C}} \left( \|\boldsymbol{\mu}_c\|_\infty - \sum_{k \in \mathcal{K} \cap \mathcal{I}^\star} \Delta_{k,c} w_{k,c}(t) \right)$$

Hence:

$$f(\underline{\mu}, \underline{\mathbf{w}}^\star) - f(\underline{\mu}, \underline{\mathbf{w}}(t)) = \sum_{c \in \mathcal{C}} \sum_{k \in \mathcal{K} \cap \mathcal{I}^\star} \Delta_{k,c} \left( w_{k,c}(t) - w_{k,c} \right)$$

$$= \sum_{k \in \mathcal{K} \cap \mathcal{I}^\star} \sum_{c \in \mathcal{C}} \frac{\Delta_{k,c}}{\mu_{k,c}} \mu_{k,c} \left( w_{k,c}(t) - w_{k,c} \right)$$

$$\leq \sigma \sum_{k \in \mathcal{K} \cap \mathcal{I}^\star} \sum_{c \in \mathcal{C}} \mu_{k,c} \left( w_{k,c}(t) - w_{k,c} \right)$$

$$\leq \sigma \sum_{k \in \mathcal{K} \cap \mathcal{I}^\star} g_k(\underline{\mu}, \underline{\mathbf{w}}(t)) - g_k(\underline{\mu}, \underline{\mathbf{w}}^\star)$$

Thus w.h.p $1 - 1/t$, we get:

$$f(\underline{\mu}, \underline{\mathbf{w}}^\star) - f(\underline{\mu}, \underline{\mathbf{w}}(t)) \overset{(19)}{\leq} \sigma \sum_{k \in \mathcal{K} \cap \mathcal{I}^\star} \rho_k(\underline{\epsilon}(t), \underline{\mathbf{w}}(t))$$

Taking the expectation:

$$\mathbb{E}\left[\mathcal{A}_2\right] \leq \sigma \mathbb{E}\left[\sum_{t=1}^{T} \sum_{k \in \mathcal{K} \cap \mathcal{I}^\star} \rho_k(\underline{\epsilon}(t), \underline{\mathbf{w}}(t))\right] + \sum_{t=1}^{T} \frac{\mu}{t}$$

$$\overset{\text{same as Eq (18)}}{\leq} \mathcal{O}\left(\sqrt{|\mathcal{C}| \cdot |\mathcal{K} \cap \mathcal{I}^\star|\log\left(T\right)T} + \mu\log\left(T\right)\right)$$

$\square$

**Upper-bounding $\mathcal{A}_3$.** This term corresponds to the rounds where step 6 of **OPLP** is feasible, but the algorithm saturates the wrong set of constraints, i.e., $\mathcal{I}_t \neq \mathcal{I}^\star$.

**Proposition E.6.** *Under* **OPLP***, the following holds:*

*1. If $\mathcal{I}_t \neq \mathcal{I}^\star$ and $\overline{\mathcal{E}(t)}$ holds, then w.h.p. $1 - \frac{1}{t}$:*
$$\rho(\boldsymbol{\epsilon}(t), \underline{\boldsymbol{w}}(t)) \geq \rho^\star.$$

*2. Moreover, it holds that*
$$\mathbb{E}[\mathcal{A}_3] \leq \mathcal{O}\left(\frac{\kappa^2}{\rho^{\star 2}} \log(T)^2\right).$$

*Proof of Proposition E.6.* We prove each point of the Proposition separately.

**Proof of Point 1.** Recall the definition of set $\Psi$ already introduced in Definition 4:
$$\psi(s, \mathcal{I}) = \left\{ \underline{\boldsymbol{w}} \in \pi_K^{|\mathcal{C}|} : \begin{array}{ll} \forall k \in \mathcal{K}, & g_k(\boldsymbol{\mu}, \underline{\boldsymbol{w}}) \geq \lambda_k - s, \\ \forall k \in \mathcal{K} \cap \mathcal{I}, & g_k(\boldsymbol{\mu}, \underline{\boldsymbol{w}}) \leq \lambda_k + s, \\ \forall (k,c) \in \mathcal{J} \cap \mathcal{I}, & h_k(c, \underline{\boldsymbol{w}}) = 0 \end{array} \right\}.$$

It is straightforward to verify that w.h.p $1 - \frac{1}{t}$:
$$\underline{\boldsymbol{w}}(t) \in \psi\left(\rho(\boldsymbol{\epsilon}(t), \underline{\boldsymbol{w}}(t)), \mathcal{I}_t\right). \tag{20}$$

**Infeasibility.** Given that $\underline{\boldsymbol{w}}(t) \in \psi\left(\rho(\boldsymbol{\epsilon}(t), \underline{\boldsymbol{w}}(t)), \mathcal{I}_t\right)$ then the latter is not empty, implying that $\rho(\boldsymbol{\epsilon}(t), \underline{\boldsymbol{w}}(t)) \geq s(\mathcal{I}_t)$.

**Performance Gap.** Recall the set $\Phi$ introduced in Definition 2:

$$\Phi(s) = \left\{ \underline{\boldsymbol{w}} \in \pi_K^{|\mathcal{C}|} : \forall k \in [K], \quad g_k(\boldsymbol{\mu}, \underline{\boldsymbol{w}}) \geq \lambda_k + s \right\}.$$

It is clear that $\forall s \in [0, \gamma]$, the set $\Phi(s)$ is non-empty. Furthermore, using Definition 3 of $S_{\gamma^\star}$, with $s_2 = \rho(\boldsymbol{\epsilon}(t), \underline{\boldsymbol{w}}(t)) \leq \gamma$ and $s_1 = 0$ yields:
$$\max_{\underline{\boldsymbol{w}} \in \Phi(0)} f(\boldsymbol{\mu}, \underline{\boldsymbol{w}}) - \max_{\underline{\boldsymbol{w}} \in \Phi(\rho(\boldsymbol{\epsilon}(t), \underline{\boldsymbol{w}}(t)))} f(\boldsymbol{\mu}, \underline{\boldsymbol{w}}) \leq S_{\gamma^\star} \rho(\boldsymbol{\epsilon}(t), \underline{\boldsymbol{w}}(t)),$$

and hence:
$$f(\boldsymbol{\mu}, \underline{\boldsymbol{w}}^\star) - \max_{\underline{\boldsymbol{w}} \in \Phi(\rho(\boldsymbol{\epsilon}(t), \underline{\boldsymbol{w}}(t)))} f(\boldsymbol{\mu}, \underline{\boldsymbol{w}}) \leq S_{\gamma^\star} \rho(\boldsymbol{\epsilon}(t), \underline{\boldsymbol{w}}(t))$$
$$\implies f(\boldsymbol{\mu}, \underline{\boldsymbol{w}}^\star) - \mathbf{LP}(\boldsymbol{\mu}, \mathbf{LCB}(t)) \leq S_{\gamma^\star} \rho(\boldsymbol{\epsilon}(t), \underline{\boldsymbol{w}}(t))$$
$$\implies f(\boldsymbol{\mu}, \underline{\boldsymbol{w}}^\star) - \mathbf{LP}(\boldsymbol{\mu}, \mathbf{LCB}(t)) \leq \max(1, S_{\gamma^\star}) \rho(\boldsymbol{\epsilon}(t), \underline{\boldsymbol{w}}(t)).$$

Now using Definition 5:
$$\max_{\underline{\boldsymbol{w}} \in \psi(\rho(\boldsymbol{\epsilon}(t), \underline{\boldsymbol{w}}(t)), \mathcal{I}_t)} f(\boldsymbol{\mu}, \underline{\boldsymbol{w}}) - \max_{\underline{\boldsymbol{w}} \in \psi(s(\mathcal{I}_t), \mathcal{I}_t)} f(\boldsymbol{\mu}, \underline{\boldsymbol{w}}) \leq \mathcal{L}(\mathcal{I}_t) \left(\rho(\boldsymbol{\epsilon}(t), \underline{\boldsymbol{w}}(t)) - s(\mathcal{I}_t)\right)$$
$$\implies \mathbf{LP}(\boldsymbol{\mu}, \mathbf{LCB}(t)) - \max_{\underline{\boldsymbol{w}} \in \psi(s(\mathcal{I}_t), \mathcal{I}_t)} f(\boldsymbol{\mu}, \underline{\boldsymbol{w}}) \leq \mathcal{L}(\mathcal{I}_t) \left(\rho(\boldsymbol{\epsilon}(t), \underline{\boldsymbol{w}}(t)) - s(\mathcal{I}_t)\right).$$

We conclude that:
$$f(\boldsymbol{\mu}, \underline{\boldsymbol{w}}^\star) - \max_{\underline{\boldsymbol{w}} \in \psi(s(\mathcal{I}_t), \mathcal{I}_t)} f(\boldsymbol{\mu}, \underline{\boldsymbol{w}}) + \mathcal{L}(\mathcal{I}_t) s(\mathcal{I}_t) \leq (\max(1, S_{\gamma^\star}) + \mathcal{L}(\mathcal{I}_t)) \rho(\boldsymbol{\epsilon}(t), \underline{\boldsymbol{w}}(t))$$
$$\implies \frac{f(\boldsymbol{\mu}, \underline{\boldsymbol{w}}^\star) - \max\limits_{\underline{\boldsymbol{w}} \in \psi(0, \mathcal{I}_t)} f(\boldsymbol{\mu}, \underline{\boldsymbol{w}}) + \mathcal{L}(\mathcal{I}_t) s(\mathcal{I}_t)}{\max(1, S_{\gamma^\star}) + \mathcal{L}(\mathcal{I}_t)} \leq \rho(\boldsymbol{\epsilon}(t), \underline{\boldsymbol{w}}(t))$$
$$\implies \frac{f(\boldsymbol{\mu}, \underline{\boldsymbol{w}}^\star) - \max\limits_{\underline{\boldsymbol{w}} \in \psi(0, \mathcal{I}_t)} f(\boldsymbol{\mu}, \underline{\boldsymbol{w}})}{\max(1, S_{\gamma^\star}) + \mathcal{L}(\mathcal{I}_t)} \leq \rho(\boldsymbol{\epsilon}(t), \underline{\boldsymbol{w}}(t))$$
$$\implies \rho(\mathcal{I}_t) \leq \rho(\boldsymbol{\epsilon}(t), \underline{\boldsymbol{w}}(t)).$$

Hence, if $\mathcal{I}_t$ is sub-optimal then $\rho(\mathcal{I}_t) \geq \rho^\star$ which concludes the proof of first point.

**Proof of Point 2.**

$$\mathcal{A}_3 = \sum_{t=1}^{T} \left( f(\underline{\boldsymbol{\mu}}, \underline{\boldsymbol{w}}^\star) - f(\underline{\boldsymbol{\mu}}, \underline{\boldsymbol{w}}(t)) \right)_+ \mathbb{1}_{[\mathcal{I}_t \neq \mathcal{I}^\star]} \mathbb{1}_{\overline{[\mathcal{E}(t)]}}$$

$$\leq \mu \sum_{t=1}^{T} \mathbb{1}_{[\mathcal{I}_t \neq \mathcal{I}^\star]} \mathbb{1}_{\overline{[\mathcal{E}(t)]}}$$

$$\implies \mathbb{E}[\mathcal{A}_3] \leq \mu \sum_{t=1}^{T} \mathbb{E} \left[ \mathbb{1}_{[\mathcal{I}_t \neq \mathcal{I}^\star]} \mathbb{1}_{\overline{[\mathcal{E}(t)]}} \right]$$

$$\leq \mu \sum_{t=1}^{T} \mathbb{E} \left[ \mathbb{1}_{[\mathcal{I}_t \neq \mathcal{I}^\star]} \mathbb{1}_{\overline{[\mathcal{E}(t)]}} \right]$$

$$\leq \mu \sum_{t=1}^{T} \mathbb{E} \left[ \mathbb{1}_{[\mathcal{I}_t \neq \mathcal{I}^\star]} \mathbb{1}_{\overline{[\mathcal{E}(t)]}} \mid \mathcal{GE} \right] . 1 + \sum_{t=1}^{T} \frac{\mu}{t}$$

$$\leq \mu \sum_{t=1}^{T} \mathbb{E} \left[ \mathbb{1}_{[\rho(\underline{\boldsymbol{\epsilon}}(t), \underline{\boldsymbol{w}}(t)) \geq \rho^\star]} \mathbb{1}_{\overline{[\mathcal{E}(t)]}} \mid \mathcal{GE} \right] + \mathcal{O}(\mu \log(T))$$

$$\leq \mu \sum_{t=1}^{T} \mathbb{E} \left[ \frac{\rho(\underline{\boldsymbol{\epsilon}}(t), \underline{\boldsymbol{w}}(t))^2}{\rho^{\star 2}} \right] + \mathcal{O}(\mu \log(T))$$

$$\overset{\text{same as Eq (16)}}{\leq} \mathcal{O} \left( \frac{\kappa^2}{\rho^{\star 2}} \log(T)^2 \right).$$

$\square$

In conclusion, combining Proposition E.4, Proposition E.5, and Proposition E.6 completes the proof of the upper bound on the regret of **OPLP**.

## F    LOWER BOUND

### F.1    LOWER BOUND THEOREM

**Theorem 5.3** (**Lower Bound**). *Let $\boldsymbol{\nu}^{(0)}$ and $\Upsilon(\boldsymbol{\nu}^{(0)}, \varepsilon)$ defined in Table 1 and Eq. 3, then:*

*(i) For $T \geq 16$, there exists $\varepsilon_T$ small enough such that:*

$$\min_{\pi} \max_{\boldsymbol{\nu} \in \Upsilon(\boldsymbol{\nu}^{(0)}, \varepsilon_T)} \mathbb{E} \left[ \mathcal{R}_{\boldsymbol{\nu}, \pi}(T) + \mathcal{V}_{\boldsymbol{\nu}, \pi}(T) \right] = \Omega \left( \sqrt{T} \right).$$

*(ii) For any consistent policy $\pi$, $\exists\, T_0 \geq 0$ s.t. $\forall\, T \geq T_0$,   $\mathbb{E} \left[ \mathcal{R}_{\boldsymbol{\nu}^{(0)}, \pi}(T) \right] = \Omega \left( \log T \right).$*

*Proof of Theorem 5.3.* We proof each point separately.

**(i) First Lower Bound.**    We start by proving the first lower bound focusing on the sum of the regret and constraints violation.

**1. Used Instances.**    Consider the nominal instance $\boldsymbol{\nu}^{(0)}$ and two perturbed instances, $\boldsymbol{\nu}_+$ and $\boldsymbol{\nu}_-$, both belonging to the uncertainty set $\Upsilon(\boldsymbol{\nu}^{(0)}, \varepsilon)$ with Gaussian distributions $\mathcal{N}(., 1)$. These instances are respectively defined in Table 2 and Table 3, with the perturbation parameter $\varepsilon$ constrained to the interval $\left(0, \frac{1}{4}\right)$. Each of these instances admits a distinct optimal allocation, summarized in Table 4 and Table 5 respectively.

**2. Effect of Wrong Beliefs on the Regret–Constraint Violations Trade-off.**    The lower bound is derived from the fact that an incorrect belief about the ground truth leads to either non-zero regret when the belief is overly pessimistic, or a constraint violation when the belief is overly optimistic.

Let $w_{2,2}^0 = 1/2$ be the optimal allocation under the nominal instance $\boldsymbol{\nu}^0$ of the second arm at the secon context:

| Arm $k$ | $p_c \, \mu_{k,c}$ | | | $\lambda$ |
|---|---|---|---|---|
| | $c=1$ | $c=2$ | $c=3$ | |
| k=1 | $\mu_{1,1}=3$ | $\mu_{1,2}=1$ | $\mu_{1,3}=1$ | $1$ |
| k=2 | $\mu_{2,1}=0$ | $\mu_{2,2}=\frac{1+\varepsilon}{2}$ | $\mu_{2,3}=0$ | $\frac{1}{4}$ |
| k=3 | $\mu_{3,1}=0$ | $\mu_{3,2}=0$ | $\mu_{3,3}=2$ | $1$ |

Table 2: $\boldsymbol{\nu}_+$ instance.

| Arm $k$ | $p_c \, \mu_{k,c}$ | | | $\lambda$ |
|---|---|---|---|---|
| | $c=1$ | $c=2$ | $c=3$ | |
| k=1 | $\mu_{1,1}=3$ | $\mu_{1,2}=1$ | $\mu_{1,3}=1$ | $1$ |
| k=2 | $\mu_{2,1}=0$ | $\mu_{2,2}=\frac{1-\varepsilon}{2}$ | $\mu_{2,3}=0$ | $\frac{1}{4}$ |
| k=3 | $\mu_{3,1}=0$ | $\mu_{3,2}=0$ | $\mu_{3,3}=2$ | $1$ |

Table 3: $\boldsymbol{\nu}_-$ instance.

| Arm $k$ | $\underline{\boldsymbol{w}}^\star$ | | | $\lambda$ |
|---|---|---|---|---|
| | $c=1$ | $c=2$ | $c=3$ | |
| 1 | $1$ | $\frac{1+2\varepsilon}{2(1+\varepsilon)}$ | $0$ | $1$ |
| 2 | $0$ | $\frac{1}{2(1+\varepsilon)}$ | $0$ | $\frac{1}{4}$ |
| 3 | $0$ | $0$ | $1$ | $1$ |

Table 4: Optimal allocation for instance $\boldsymbol{\nu}_+$.

| Arm $k$ | $\underline{\boldsymbol{w}}^\star$ | | | $\lambda$ |
|---|---|---|---|---|
| | $c=1$ | $c=2$ | $c=3$ | |
| 1 | $1$ | $\frac{1-2\varepsilon}{2(1-\varepsilon)}$ | $0$ | $1$ |
| 2 | $0$ | $\frac{1}{2(1-\varepsilon)}$ | $0$ | $\frac{1}{4}$ |
| 3 | $0$ | $0$ | $1$ | $1$ |

Table 5: Optimal allocation for instance $\boldsymbol{\nu}_-$.

- If the instance is $\boldsymbol{\nu}_+$ and $w_{2,2}(t) \geq w_{2,2}^0$, then the algorithm leads to a regret of at least

$$r(t) \geq \left( w_{2,2}^0 - \frac{1}{2(1+\varepsilon)} \right) \left( 1 - \frac{1+\varepsilon}{2} \right) \geq \left( \frac{1}{2} - \frac{1}{2(1+\varepsilon)} \right) \cdot \frac{1-\varepsilon}{2} = \frac{\varepsilon(1-\varepsilon)}{4(1+\varepsilon)} \geq \frac{\varepsilon}{10}.$$

- If the instance is $\boldsymbol{\nu}_-$ and $w_{2,2}(t) \leq w_{2,2}^0$, then the algorithm leads to a constraint violation of at least

$$v(t) \geq \frac{1}{4} - \frac{1-\varepsilon}{2} w_{2,2}^0 \geq \frac{1}{4} - \frac{1-\varepsilon}{4} \geq \frac{\varepsilon}{10}$$

**3. Information Theory.** Let $\mathbb{P}_{\boldsymbol{\nu}_+}$ and $\mathbb{P}_{\boldsymbol{\nu}_-}$ denote the distributions induced by the learning algorithm under the two problem instances $\boldsymbol{\nu}_+$ and $\boldsymbol{\nu}_-$, respectively. Explicitly consider the policy $\pi$, and let $\mathcal{F}_T$ denote the trajectory induced by that policy. Given the assumptions on the rewards, the KL-divergence between the distributions over the trajectory of $T$ rounds satisfies:

$$D\left( \mathbb{P}_{\boldsymbol{\nu}_+}(\mathcal{F}_T) \,\|\, \mathbb{P}_{\boldsymbol{\nu}_-}(\mathcal{F}_T) \right) \leq \frac{T\varepsilon^2}{2} \tag{21}$$

*Proof of equation* (21). We denote by $G_\pi(t)$ the aggregated gain received by the player at time $t$ under policy $\pi$. Hence:

$$\begin{aligned}
D\left( \mathbb{P}_{\boldsymbol{\nu}_+}(G_\pi(t)) \,\|\, \mathbb{P}_{\boldsymbol{\nu}_-}(G_\pi(t)) \right) &= \sum_{c=1}^{3}\sum_{k=1}^{3} w_{k,c}^\pi(t) D\left( \mathcal{N}(\mu_{k,c}^{\boldsymbol{\nu}_+}, 1) \,\|\, \mathcal{N}(\mu_{k,c}^{\boldsymbol{\nu}_-}, 1) \right) \\
&= w_{2,2}^\pi(t) D\left( \mathcal{N}(\mu_{2,2}^{\boldsymbol{\nu}_+}, 1) \,\|\, \mathcal{N}(\mu_{2,2}^{\boldsymbol{\nu}_-}, 1) \right) \\
&\leq \frac{\varepsilon^2}{2}
\end{aligned}$$

Summing over the trajectory ends the proof. $\qquad\square$

Consider the event

$$\Gamma = \left\{ \sum_{t=1}^{T} \mathbb{I}\left\{ w_{2,2}^\pi(t) \geq w_{2,2}^0 \right\} \geq \frac{T}{2} \right\}.$$

Note that:

$$\Gamma \text{ holds under } \boldsymbol{\nu}_+ \quad \Longrightarrow \quad \mathcal{R}_T \geq \frac{\varepsilon T}{10},$$

and similarly,

$$\bar{\Gamma} \text{ holds under } \boldsymbol{\nu}_- \quad \Longrightarrow \quad \mathcal{V}_T \geq \frac{\varepsilon T}{10}.$$

Using the Bretagnolle-Huber inequality Lattimore and Szepesvári (2020):

$$\mathbb{P}_{\boldsymbol{\nu}_+}(\Gamma) + \mathbb{P}_{\boldsymbol{\nu}_-}(\bar{\Gamma}) \geq \frac{1}{2}\exp\left(-D\left(\mathbb{P}_{\boldsymbol{\nu}_+}(\mathcal{F}_T) \,\|\, \mathbb{P}_{\boldsymbol{\nu}_-}(\mathcal{F}_T)\right)\right) \geq \frac{1}{2}\exp\left(-\frac{T\varepsilon^2}{2}\right)$$

**4. Lower Bound Explicitly.** For any policy $\pi$ generated by any learning algorithm:

$$\mathbb{E}\left[\mathcal{R}_{\boldsymbol{\nu}_+,\pi}(T) + \mathcal{V}_{\boldsymbol{\nu}_+,\pi}(T)\right] + \mathbb{E}\left[\mathcal{R}_{\boldsymbol{\nu}_-,\pi}(T) + \mathcal{V}_{\boldsymbol{\nu}_-,\pi}(T)\right] \geq \mathbb{E}\left[\mathcal{R}_{\boldsymbol{\nu}_+,\pi}(T)\mathbb{1}_{[\Gamma]}\right] + \mathbb{E}\left[\mathcal{V}_{\boldsymbol{\nu}_-,\pi}(T)\mathbb{1}_{[\bar{\Gamma}]}\right]$$

$$\geq \frac{T\varepsilon}{10}\left(\mathbb{P}_{\boldsymbol{\nu}_+}(\Gamma) + \mathbb{P}_{\boldsymbol{\nu}_-}(\bar{\Gamma})\right)$$

$$\geq \frac{T\varepsilon}{20}\exp\left(-\frac{T\varepsilon^2}{2}\right)$$

Choosing $\varepsilon = \frac{1}{\sqrt{T}}$ for $T \geq 16$ leads to:

$$\mathbb{E}\left[\mathcal{R}_{\boldsymbol{\nu}_+,\pi}(T) + \mathcal{V}_{\boldsymbol{\nu}_+,\pi}(T)\right] + \mathbb{E}\left[\mathcal{R}_{\boldsymbol{\nu}_-,\pi}(T) + \mathcal{V}_{\boldsymbol{\nu}_-,\pi}(T)\right] \geq \frac{\sqrt{T}}{20e^2}$$

Let $\varepsilon_T = T^{-1/2}$ (where $T \geq 16$). For any policy $\pi$, since both $\boldsymbol{\nu}_+$ and $\boldsymbol{\nu}_-$ belong to $\Upsilon(\boldsymbol{\nu}^{(0)}, \varepsilon_T)$, the following holds:

$$\max_{\boldsymbol{\nu}\in\Upsilon(\boldsymbol{\nu}^{(0)},\varepsilon_T)} \mathbb{E}\left[\mathcal{R}_{\boldsymbol{\nu},\pi}(T) + \mathcal{V}_{\boldsymbol{\nu},\pi}(T)\right] \geq \frac{1}{2}\left(\mathbb{E}\left[\mathcal{R}_{\boldsymbol{\nu}_+,\pi}(T) + \mathcal{V}_{\boldsymbol{\nu}_+,\pi}(T)\right] + \mathbb{E}\left[\mathcal{R}_{\boldsymbol{\nu}_-,\pi}(T) + \mathcal{V}_{\boldsymbol{\nu}_-,\pi}(T)\right]\right)$$

$$\implies \max_{\boldsymbol{\nu}\in\Upsilon(\boldsymbol{\nu}^{(0)},\varepsilon_T)} \mathbb{E}\left[\mathcal{R}_{\boldsymbol{\nu},\pi}(T) + \mathcal{V}_{\boldsymbol{\nu},\pi}(T)\right] \geq \frac{\sqrt{T}}{40e^2}$$

$$\implies \min_{\pi}\max_{\boldsymbol{\nu}\in\Upsilon(\boldsymbol{\nu}^{(0)},\varepsilon_T)} \mathbb{E}\left[\mathcal{R}_{\boldsymbol{\nu},\pi}(T) + \mathcal{V}_{\boldsymbol{\nu},\pi}(T)\right] \geq \frac{\sqrt{T}}{40e^2}.$$

This concludes the proof of (i).

**This instance is not a corner case.** It is worth noting that the instance used in the lower bound is not a corner case; that is, the characterizing gaps are non-zero. In particular, strict feasibility is ensured by setting $\gamma^\star = \frac{1}{8}$, and the performance gap $\rho^\star > 0$ because the problem is not degenerate. Specifically, there exists a unique solution, which results in a non-zero performance gap while activating suboptimal indices.

**(ii) Second Lower Bound.** We now derive the second lower bound on the regret.

**Used Instance.** Consider the same nominal instance $\boldsymbol{\nu}^{(0)}$ as previously defined, and introduce a modified instance $\boldsymbol{\nu}'$ that differs from $\boldsymbol{\nu}^{(0)}$ only in the reward parameter of arm 1 in the third context: $\mu_{1,3}(\boldsymbol{\nu}') = 2 + \varepsilon'$, $\varepsilon' \in (0, 1]$. The modified instance $\boldsymbol{\nu}'$ and its corresponding optimal allocations are summarized in Table 6 and Table 7, respectively.

| Arm $k$ | $p_c\,\mu_{k,c}$ | | | $\lambda$ |
|---|---|---|---|---|
| | $c=1$ | $c=2$ | $c=3$ | |
| 1 | $\mu_{1,1}=3$ | $\mu_{1,2}=1$ | $\mu_{1,3}=2+\varepsilon'$ | 1 |
| 2 | $\mu_{2,1}=0$ | $\mu_{2,2}=\frac{1}{2}$ | $\mu_{2,3}=0$ | $\frac{1}{4}$ |
| 3 | $\mu_{3,1}=0$ | $\mu_{3,2}=0$ | $\mu_{3,3}=2$ | 1 |

Table 6: Instance $\boldsymbol{\nu}'$.

| Arm $k$ | $\boldsymbol{w}^\star$ | | | $\lambda$ |
|---|---|---|---|---|
| | $c=1$ | $c=2$ | $c=3$ | |
| 1 | 1 | $\frac{1}{2}$ | $\frac{1}{2}$ | 1 |
| 2 | 0 | $\frac{1}{2}$ | 0 | $\frac{1}{4}$ |
| 3 | 0 | 0 | $\frac{1}{2}$ | 1 |

Table 7: Optimal allocation for instance $\boldsymbol{\nu}'$.

Note that context $c = 3$ does not contribute to satisfying the constraint of arm $k = 1$, while under $\boldsymbol{\nu}^{(0)}$ arm $k = 3$ is the best-performing arm in that context and can only satisfy its constraint due to rewards obtained in context $c = 3$.

Let $\pi$ be any consistent policy such that there exists a strictly positive constant $\beta$ such that for sufficiently large $T$

$$\mathbb{E}[\mathcal{R}_{\boldsymbol{\nu}^{(0)},\pi}(T) + \mathcal{R}_{\boldsymbol{\nu}',\pi}(T)] \leq \beta\sqrt{T}. \tag{22}$$

**Information Theory.** Consider the event

$$\Gamma' = \left\{ n_{1,3}(T) \geq \frac{p_3 T}{4} \right\}.$$

Note that:

$$\Gamma' \text{ holds under } \boldsymbol{\nu}^{(0)} \implies \mathcal{R}_{\boldsymbol{\nu}^{(0)},\pi}(T) \geq \frac{(2-1)T}{4} = \frac{T}{4},$$

$$\bar{\Gamma}' \text{ holds under } \boldsymbol{\nu}' \implies \mathcal{R}_{\boldsymbol{\nu}',\pi}(T) \geq \varepsilon'(\frac{T}{2} - \frac{T}{4}) \geq \frac{\varepsilon' T}{4}.$$

Using the Bretagnolle-Huber inequality Lattimore and Szepesvári (2020):

$$\mathbb{P}_{\boldsymbol{\nu}^{(0)}}(\Gamma') + \mathbb{P}_{\boldsymbol{\nu}'}(\bar{\Gamma}') \geq \frac{1}{2} \exp\left(-D\left(\mathbb{P}_{\boldsymbol{\nu}^{(0)}}(\mathcal{F}_T) \,\|\, \mathbb{P}_{\boldsymbol{\nu}'}(\mathcal{F}_T)\right)\right) \geq \frac{1}{2} \exp\left(-\frac{\mathbb{E}_{\boldsymbol{\nu}^{(0)},\pi}[n_{1,3}(T)](1+\varepsilon')^2}{2}\right)$$

(23)

**Lower Bound Explicitly.** Leveraging Equation (23):

$$\mathbb{E}\left[\mathcal{R}_{\boldsymbol{\nu}^{(0)},\pi}(T) + \mathcal{R}_{\boldsymbol{\nu}',\pi}(T)\right] \geq \mathbb{E}\left[\mathcal{R}_{\boldsymbol{\nu}^{(0)},\pi}(T)\mathbb{1}_{[\Gamma']}\right] + \mathbb{E}\left[\mathcal{R}_{\boldsymbol{\nu}',\pi}(T)\mathbb{1}_{[\bar{\Gamma}']}\right]$$

$$\geq \frac{T\varepsilon'}{4}\left(\mathbb{P}_{\boldsymbol{\nu}^{(0)}}(\Gamma') + \mathbb{P}_{\boldsymbol{\nu}'}(\bar{\Gamma}')\right)$$

$$\geq \frac{T\varepsilon'}{8}\exp\left(-\frac{\mathbb{E}_{\boldsymbol{\nu}^{(0)},\pi}[n_{1,3}(T)](1+\varepsilon')^2}{2}\right)$$

$$\implies \mathbb{E}_{\boldsymbol{\nu}^{(0)},\pi}[n_{1,3}(T)] \geq \frac{2}{(1+\varepsilon')^2}\left(\log\left(\frac{T\varepsilon'}{8}\right) - \log\left(\mathbb{E}\left[\mathcal{R}_{\boldsymbol{\nu}^{(0)},\pi}(T) + \mathcal{R}_{\boldsymbol{\nu}',\pi}(T)\right]\right)\right)$$

$$\overset{Eq.(22)}{\geq} \frac{2}{(1+\varepsilon')^2}\left(\log\left(\frac{T\varepsilon'}{8}\right) - \log\left(\beta\sqrt{T}\right)\right)$$

$$\geq \frac{2}{(1+\varepsilon')^2}\left(\frac{1}{2}\log(T) + \log\left(\frac{\varepsilon'}{8\beta}\right)\right)$$

Consequently, for all $T \geq T_0$, where $T_0$ is defined by the condition $\frac{1}{2}\log(T_0) \gg |\log(\varepsilon'/(8\beta))|$, we obtain the following regret lower bound:

$$\mathbb{E}\left[\mathcal{R}_{\boldsymbol{\nu}^{(0)},\pi}(T)\right] \geq (2-1)\mathbb{E}_{\boldsymbol{\nu}^{(0)},\pi}[n_{1,3}(T)]$$

$$= \Omega\left(\log(T)\right).$$

$\square$

### F.2 RICH FAMILY OF MAB-ARC WITH NO FREE EXPLORATION

**Lemma 5.1.** *For any **MAB-ARC** instance such that $K > 2$ and $|\mathcal{C}| > 1$, there exists at least one pair $(k,c) \in \mathcal{J}$ for which the optimal allocation satisfies $w^\star_{k,c} = 0$.*

*Proof of Lemma 5.1.* We proceed by contradiction. Suppose there exists a feasible instance such that $|\mathcal{C}| > \frac{K}{K-1}$, and for every $(k,c) \in \mathcal{J}$, it holds that $w^\star_{k,c} \neq 0$.

By Assumption 3, the linear program is feasible and non-degenerate. Hence, the optimal solution must saturate exactly $\kappa$ constraints.

There are $|\mathcal{C}|$ equality constraints (i.e., the $q_c$ constraints). Thus, $|\mathcal{C}|$ constraints are already saturated. The remaining $\kappa - |\mathcal{C}| = |\mathcal{C}|(K-1)$ constraints must be saturated by the arm-level aggregated reward constraints (i.e., the $g_k$ constraints) because by assumption, no variable $w^\star_{k,c}$ is zero, meaning that none of the non-negativity constraints $w_{k,c} \geq 0$ is active, and hence no saturation occurs there. This implies that all $|\mathcal{C}|(K-1)$ yet to saturate constraints must come from the $K$ minimum reward constraints, which is only possible if $K \geq |\mathcal{C}|(K-1)$. But this contradicts the assumption that

$|\mathcal{C}| > \frac{K}{K-1}$. Therefore there must exist at least one pair $(k,c)$ such that $w^\star_{k,c} = 0$. Given that the function $x \mapsto \frac{x}{x-1}$ is strictly decreasing for all $x \geq 3$, we obtain the inclusion relationship

$$\{\textbf{MAB-ARC}: K > 2, |\mathcal{C}| > 1\} \subset \{\textbf{MAB-ARC}: |\mathcal{C}| > \tfrac{K}{K-1}\}.$$

This completes the proof.

$\square$

## G  NUMERICAL ILLUSTRATIONS

We validate our theoretical results through numerical experiments on simulated data. Specifically, we consider the instance $\nu$ defined in Table 8, where rewards follow Gaussian distributions $\mathcal{N}(.,1)$ and contexts are uniformly distributed. The corresponding optimal allocation, which serves as our benchmark, is provided in Table 9. For this instance the optimal set of active constraints is:

$$\mathcal{I}^\star = \{2, 3, (2,1), (3,1), (3,2), (2,3)\}.$$

For comparison, we evaluate both our algorithms (**OLP** and **OPLP**) alongside **Optimistic**[3] from Guo et al. (2025) and the **DOC** and **SPOC** algorithms from Baudry et al. (2024). There is no established baseline in the literature that directly addresses contextual multi-armed bandits with revenue constraints for benchmarking our algorithms. However, Guo et al. (2025) introduced **Optimistic**[3] for MABs with general stochastic constraints, which can be readily adapted to our setting. In addition, one may adapt non-contextual algorithms such as **DOC** and **SPOC** by disregarding contextual information. While this leads to an unfair comparison—since the algorithms operate under different informational assumptions—it underscores the importance of leveraging contextual information when available, as doing so yields markedly superior performance. Indeed, **DOC** and **SPOC** inherently neglect the contextual dimension of the problem, instead estimating quantities of the form $\lambda_k / \sum_{c \in \mathcal{C}} p_c \mu_{k,c}$ for each arm and proceeding accordingly.

We conduct experiments over $T = 50 \times 10^3$ rounds, repeated for 5 independent epochs.

| Arm $k$ | $p_c\,\mu_{k,c}$ | | | $\lambda$ |
|---|---|---|---|---|
| | $c=1$ | $c=2$ | $c=3$ | |
| 1 | $\mu_{1,1}=3$ | $\mu_{1,2}=1$ | $\mu_{1,3}=2$ | $1$ |
| 2 | $\mu_{2,1}=0$ | $\mu_{2,2}=\frac{1}{2}$ | $\mu_{2,3}=0$ | $\frac{1}{4}$ |
| 3 | $\mu_{3,1}=0$ | $\mu_{3,2}=0$ | $\mu_{3,3}=1$ | $\frac{1}{2}$ |

Table 8: Instance $\nu$.

| Arm $k$ | $\underline{\boldsymbol{w}}^\star$ | | | $\lambda$ |
|---|---|---|---|---|
| | $c=1$ | $c=2$ | $c=3$ | |
| 1 | $1$ | $\frac{1}{2}$ | $\frac{1}{2}$ | $1$ |
| 2 | $0$ | $\frac{1}{2}$ | $0$ | $\frac{1}{4}$ |
| 3 | $0$ | $0$ | $\frac{1}{2}$ | $\frac{1}{2}$ |

Table 9: Optimal allocation for instance $\nu$.

Figure 2 reports:

- The cumulative regret and constraint violation for **OLP**, **OPLP** and **Optimistic**[3].

- The cumulative performance of all five algorithms

From a regret perspective, **OLP** achieves superior performance compared to **OPLP**, exhibiting logarithmic regret versus the $\mathcal{O}(\sqrt{T})$ regret of **OPLP**. Conversely, **OPLP** ensures stronger constraint satisfaction than **OLP**, achieving logarithmic rather than $\mathcal{O}(\sqrt{T})$ constraint violation. In contrast, **Optimistic**[3] yields $\mathcal{O}(\sqrt{T})$ bounds for both regret and constraint violation. Hence, **OLP** and **OPLP** are better suited to the considered setting, as they achieve a polylogarithmic regime compared to the $\mathcal{O}(\sqrt{T})$ behavior of **Optimistic**[3].

The third plot highlights the performance advantage of our contextual approach: both **OLP** and **OPLP** outperform the non-contextual baselines **DOC** and **SPOC**, justifying the need for additional work beyond existing literature to better adapt to the **MAB-ARC** setting.

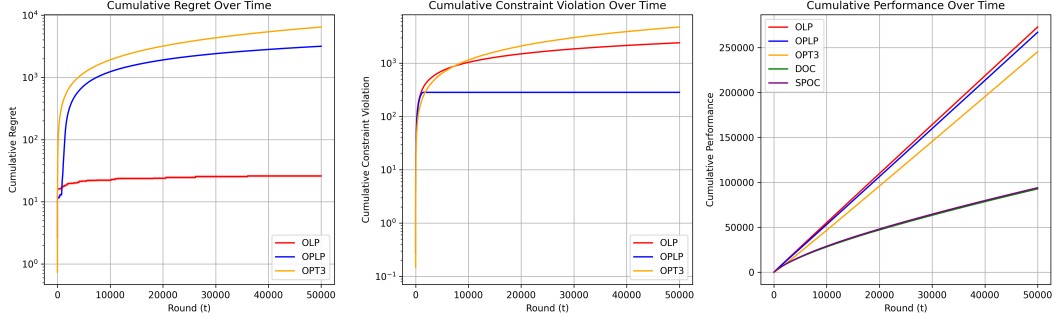

Figure 2: Display of the cumulative regret and constraint violation for **OLP**, **OPLP** and **Optimistic**[3] (denoted by OPT3), and the cumulative performance of all five algorithms, under identical conditions ($K = 3$, $|\mathcal{C}| = 3$, $T = 50{,}000$, Gaussian distributions $\mathcal{N}(.,1)$, 5 epochs).

**Non Saturating Arms.** For the sake of completeness, we include an additional experiment to illustrate the correct adaptability of our analysis in the regime where no arm saturates its revenue constraints. We consider an instance $\nu'$ defined in Table 11,where rewards follow Gaussian distributions $\mathcal{N}(.,1)$ and contexts are uniformly distributed, and, according to the oracle solution given in Table 11, none of the arms saturates its respective revenue constraint. For this instance the optimal set of active constraints is:

$$\mathcal{I}^\star = \{(2,1),(3,1),(1,2),(3,2),(1,3),(2,3)\}.$$

| Arm $k$ | $p_c\,\mu_{k,c}$ | | | $\lambda$ |
|---|---|---|---|---|
| | $c = 1$ | $c = 2$ | $c = 3$ | |
| 1 | $\mu_{1,1} = 3$ | $\mu_{1,2} = 1$ | $\mu_{1,3} = 1$ | 1 |
| 2 | $\mu_{2,1} = 0$ | $\mu_{2,2} = 3$ | $\mu_{2,3} = 1$ | 1 |
| 3 | $\mu_{3,1} = 1$ | $\mu_{3,2} = 1$ | $\mu_{3,3} = 3$ | 1 |

Table 10: Instance $\nu'$.

| Arm $k$ | $\underline{w}^\star$ | | | $\lambda$ |
|---|---|---|---|---|
| | $c = 1$ | $c = 2$ | $c = 3$ | |
| 1 | 1 | 0 | 0 | 1 |
| 2 | 0 | 1 | 0 | 1 |
| 3 | 0 | 0 | 1 | 1 |

Table 11: Optimal allocation for instance $\nu'$.

In accordance with Theorems 5.1 and 5.2, Figure 3 shows that both the regret and the constraint violations for **OLP** and **OPLP** exhibit logarithmic behavior.

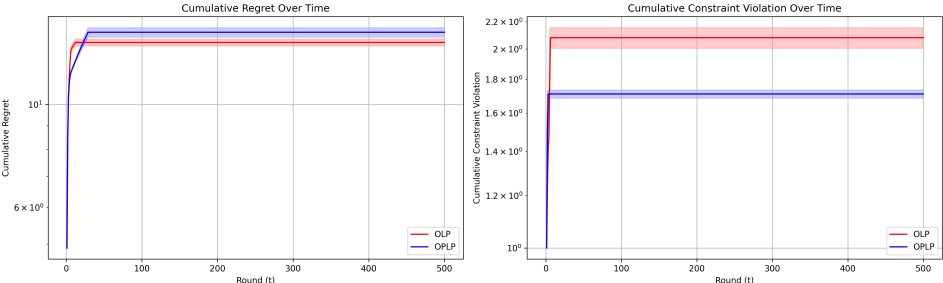

Figure 3: Cumulative regret and constraint violation for **OLP** and **OPLP**, evaluated under identical conditions ($K = 3$, $|\mathcal{C}| = 3$, $T = 500$, Gaussian distributions $\mathcal{N}(.,1)$, 5 epochs) on an instance where, according to the optimal stationary policy, no arm saturates its revenue constraint.

### G.1 SENSITIVITY OF **OPLP** TO THE FEASIBILITY MARGIN

The **OPLP** algorithm heavily relies on the feasibility margin $\gamma^\star$, as it adopts a conservative strategy that prioritizes constraint satisfaction through the use of a **LCB** estimator for the constraints. However, this approach may not always be feasible, which motivates the use of a **UCB**-based strategy as a

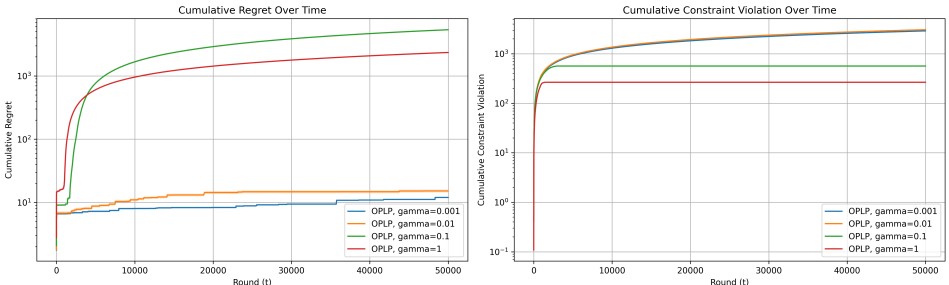

Figure 4: Cumulative regret and constraint violation of **OPLP** under different values of the feasibility margin $\gamma^\star$. ($K = 3$, $|\mathcal{C}| = 3$, $T = 50,000$, Gaussian distributions $\mathcal{N}(.,1)$, 5 epochs)

fallback. To illustrate this, we deploy **OPLP** under different values of $\gamma^\star$, as shown in Figure 4. In fact, for small values of $\gamma^\star$ (i.e., $\gamma^\star = 0.001$ and $\gamma^\star = 0.01$), the **LCB** estimator was never feasible during the entire time horizon, and the behavior of **OLP** was observed instead, yielding logarithmic regret and $\mathcal{O}(\sqrt{T})$ constraint violations. However, for larger values such as $\gamma^\star = 0.1$ and $\gamma^\star = 1$ , the **LCB** estimator becomes feasible, and the standard behavior of **OPLP**; logarithmic constraint violations and $\mathcal{O}(\sqrt{T})$ regret, is recovered. Notably, we observe the expected rate of $\frac{1}{\gamma^{\star 2}}$ scaling in front of the logarithmic behavior of the constraint violation under **OPLP**.

## H   LAZY UPDATE

---

**Lazy-OLP:** Lazy Optimistic Linear Programming

1 **Inputs:** $\{\lambda_k\}_{k \in \{1,\ldots,K\}}, \{p_c\}_{c \in \mathcal{C}}$
2 $\tau = 1$ {This is the last timestep that we changed the policy}
3 **for** $t = 1, \ldots, T$ **do**
4      Observe context $c_t$
5      Set $\delta \leftarrow 1/t$
6      **if** $\exists c, k, \; n_{k,c}(t-1) > 2n_{k,c}(\tau)$ **then**
7          $\underline{w}(t) = \arg\max_{\underline{w} \in \pi_K^{|\mathcal{C}|}} \mathbf{LP}(\underline{\mathbf{UCB}}(t), \underline{\mathbf{UCB}}(t))$
8          $\tau = t$
9      **end**
10      **else**
11          $\underline{w}(t) = \underline{w}(\tau)$
12      **end**
13      Sample arm $k_t \sim \boldsymbol{w}_{c_t}(t)$
14      Receive reward $r_t \sim \mu_{k_t, c_t}$
15      Update $\underline{\boldsymbol{n}}(t), \hat{\boldsymbol{\mu}}(t), \underline{\boldsymbol{\epsilon}}(t)$
16      Update history $\mathcal{H}_t = \mathcal{H}_{t-1} \cup \{c_t, k_t, r_t\}$
17 **end**

---

**Lazy-OPLP:** Lazy Optimistic-Pessimistic Linear Programming

1 **Inputs:** $\{\lambda_k\}_{k \in \{1,\ldots,K\}}, \{p_c\}_{c \in \mathcal{C}}$
2 $\tau = 1$ {This is the last timestep that we changed the policy}
3 **for** $t = 1, \ldots, T$ **do**
4      Observe context $c_t$
5      Set $\delta \leftarrow 1/t$
6      **if** $\exists c, k, \; n_{k,c}(t-1) > 2n_{k,c}(\tau)$ **then**
7          **if** $LP(\underline{\mathbf{UCB}}(t), \underline{\mathbf{LCB}}(t))$ *is feasible* **then**
8              $\underline{w}(t) = \arg\max_{\underline{w} \in \pi_K^{|\mathcal{C}|}} \mathbf{LP}(\underline{\mathbf{UCB}}(t), \underline{\mathbf{LCB}}(t))$
9          **end**
10          **else**
11              $\underline{w}(t) = \arg\max_{\underline{w} \in \pi_K^{|\mathcal{C}|}} \mathbf{LP}(\underline{\mathbf{UCB}}(t), \underline{\mathbf{UCB}}(t))$
12          **end**
13          $\tau = t$
14      **end**
15      **else**
16          $\underline{w}(t) = \underline{w}(\tau)$
17      **end**
18      Sample arm $k_t \sim \boldsymbol{w}_{c_t}(t)$
19      Receive reward $r_t \sim \mu_{k_t, c_t}$
20      Update $\underline{\boldsymbol{n}}(t), \hat{\boldsymbol{\mu}}(t), \underline{\boldsymbol{\epsilon}}(t)$
21      Update history $\mathcal{H}_t = \mathcal{H}_{t-1} \cup \{c_t, k_t, r_t\}$
22 **end**

---

Both **Lazy OLP** and **Lazy OPLP** employ a reduced update frequency compared to their vanilla counterparts, **OLP** and **OPLP**. The policy optimization step is triggered only when the number of pulls for any arm–context pair has doubled since the last update. Between these events, the policy and confidence intervals from the most recent update are retained. This scheme reduces the number of required computations to at most $\mathcal{O}(\log T)$ over a horizon $T$. Interestingly, this reduction in update frequency does not deteriorate the theoretical guarantees. The analysis and characterization of the saturating constraint set remain exactly valid as in the non-lazy case. Consequently, the polylogarithmic and $O(\sqrt{T})$ bounds are intrinsically tied to the respective bounds of

$$\mathbb{E}\left[\sum_{t=1}^{T}\rho_k(\boldsymbol{\epsilon}(t),\underline{\boldsymbol{w}}(t))^2\right] \quad \text{and} \quad \mathbb{E}\left[\sum_{t=1}^{T}\rho_k(\boldsymbol{\epsilon}(t),\underline{\boldsymbol{w}}(t))\right].$$

Let $\tau_t$ denote the most recent round prior to $t$ at which a policy update occurred. Under both **Lazy OLP** and **Lazy OPLP**, we have:

$$
\begin{aligned}
\mathbb{E}\left[\sum_{t=1}^{T}\rho_k(\boldsymbol{\epsilon}(t),\underline{\boldsymbol{w}}(t))^2\right] &= \mathbb{E}\left[\sum_{t=1}^{T}\sum_{c\in\mathcal{C}}w_{k,c}(t)^2\frac{2\log(2\kappa\tau_t)}{n_{k,c}(\tau_t)}\right] \\
&\leq \mathbb{E}\left[2\log(2\kappa T)\sum_{c\in\mathcal{C}}\sum_{t=1}^{T}\frac{w_{k,c}(t)^2}{n_{k,c}(\tau_t)}\right] \\
&\leq \mathbb{E}\left[2\log(2\kappa T)\sum_{c\in\mathcal{C}}\sum_{t=1}^{T}2\frac{w_{k,c}(t)^2}{n_{k,c}(t)}\right] && (n_{k,c}(t)\leq 2n_{k,c}(\tau_t)) \\
&\leq \mathcal{O}\left(\log(T)^2\right) && \text{(similar to Proposition } C.1) \\
\mathbb{E}\left[\sum_{t=1}^{T}\rho_k(\boldsymbol{\epsilon}(t),\underline{\boldsymbol{w}}(t))\right] &= \mathbb{E}\left[\sum_{t=1}^{T}\sum_{c\in\mathcal{C}}w_{k,c}(t)\sqrt{\frac{2\log(2\kappa\tau_t)}{n_{k,c}(\tau_t)}}\right] \\
&\leq \mathbb{E}\left[\sqrt{2\log(2\kappa T)}\sum_{c\in\mathcal{C}}\sum_{t=1}^{T}\frac{w_{k,c}(t)}{\sqrt{n_{k,c}(\tau_t)}}\right] \\
&\leq \mathbb{E}\left[\sqrt{2\log(2\kappa T)}\sum_{c\in\mathcal{C}}\sum_{t=1}^{T}\sqrt{2}\frac{w_{k,c}(t)}{\sqrt{n_{k,c}(t)}}\right] && (n_{k,c}(t)\leq 2n_{k,c}(\tau_t)) \\
&\leq \mathcal{O}\left(\sqrt{\log(T)T}\right) && \text{(similar to Proposition } C.1)
\end{aligned}
$$

As a result, the lazy versions of the algorithms preserve the same theoretical guarantees as their vanilla counterparts.

## I   COUNTEREXAMPLE DEMONSTRATING THE INEFFICIENCY OF GREEDY BEHAVIOR

Greedy achieves sublinear regret in the single-context setting because the constraints enforce exploration: in order to satisfy the revenue constraint for each arm, Greedy is forced to play all arms and eventually refines its estimates. However, in the multi-context setting, the constraints do not necessarily enforce exploration. For instance, consider the example in Table 12.

| Arm $k$ | $p_c\,\mu_{k,c}$ | | $\lambda$ |
|---|---|---|---|
| | $c=1$ | $c=2$ | |
| 1 | $\mu_{1,1}=1$ | $\mu_{1,2}=2$ | 0.1 |
| 2 | $\mu_{2,1}=2$ | $\mu_{2,2}=1$ | 0.1 |

Table 12: Counterexample instance illustrating Greedy's inefficiency in the multi-context setting.

In this setting, the optimal allocation is $p_{1,2}=1, p_{2,1}=1$. However, the Greedy algorithm may incorrectly conclude, with constant probability, that the optimal allocation is $p_{1,1}=1, p_{2,2}=1$,

thereby incurring linear regret. This inefficiency arises because the constraints are not tight enough to enforce sufficient exploration. It is worth noting that in the single-constraint case, Greedy may also yield linear regret if $\lambda_k = 0$ for the best arm in the instance.

