# OpenReview forum: "Contextual Multi-Armed Bandits with Minimum Aggregated Revenue Constraints"
_ICLR.cc/2026/Conference — ICLR 2026 Poster_

### Official Review · Reviewer_fiZz · 2025-10-25

**Soundness:** 3
**Presentation:** 2
**Contribution:** 3
**Rating:** 4
**Confidence:** 4

**Summary:**

This paper considers the contextual multi-arm bandit problem with minimum aggregated revenue constraints, i.e., the average revenue of each arm should be above a given threshold. This problem is subsumed by the more general problem of contextual bandits with linear constraints [Slivkins et al. 2024], but the authors go beyond their results by providing two algorithms OLP and OPLP: OLP achieves polylog regret and square root constraint violation, while OPLP has the reversed guarantee. Further, there is a lower bound regret of $\Omega(\log T)$ and a lower bound sum of regret and violation of $\Omega(\sqrt{T})$.

**Strengths:**

+ The paper studies a sub-problem of a more general problem which is of independent interest, and obtains a better upper bound result than the general problem.
+ The authors show the possibility of applying re-solving-based techniques (though not emphasized) in the contextual bandits with linear constraints, which I feel really interesting.

**Weaknesses:**

- I feel like the "contextual" term should appear in the title.
- The authors do not discuss the literature on the re-solving technique which is widely used in contextual decision-making problems, e.g., [Chen et al., NeurIPS 2024].
- Though the main results are already interesting, I still encourage the authors to justify the following points, which would be better for the understanding of the whole community:
  1. There is underlying hardness for re-solving based techniques to work in the CBwK problem. Intuitively, samples for certain (context, arm) pairs can be unattainable because the probability of selecting such a pair could be zero in the LP [Chen et al., NeurIPS 2024]. But this problem does not seem to happen for the considered problem. Can the authors explain an intuition?
  2. Why is the sub-Gaussian reward assumption necessary?
  3. Do you have a beyond-the-worst-case result, which says that the algorithm still has a performance guarantee when the non-degeneracy condition does not hold?
  4. What will happen if the fluid LP is infeasible? This may happen in real life, and is there any way to detect this without losing too much?
  5. What does $\rho^*$ mean intuitively? And can you give an intuitive explanation of why the regret relies on $\rho^*$?
- The author(s) suppose that the context prior is known. This assumption should be relaxed for a general CB paper.


Reference:

Chen et al. Contextual Decision-Making with Knapsacks Beyond the Worst Case. NeurIPS 2024.

**Questions:**

See the above "weaknesses" part. Feel free to answer those questions.

---

> ### Author Response · Authors · 2025-11-17
>
> Dear reviewer fiZz, we thank you for your review and the effort you dedicated to it. In the following, we provide detailed answers to your questions and address the points mentioned in the weaknesses section:
>
> * We thank the reviewer for the suggestion. Following this recommendation, the revised title of the paper is: **Contextual Multi-Armed Bandits with Minimum Aggregated Revenue Constraints**.
>
> * We thank you for pointing out the literature on re-solving techniques, in particular the work of [Chen et al., NeurIPS 2024]. The authors consider a contextual decision-making with knapsacks setting where a random request is revealed to the learner, who then selects an action and receives a reward while incurring a cost. Both the reward and the cost depend on the observed request, the chosen action, and an external random factor. Importantly, the associated reward and cost functions are known to the learner. Building on the network revenue management literature, the authors propose an elegant re-solving method in which, at each iteration, the distributions of the requests and external factors are estimated and an approximation of the fluid linear program is solved.  While their work and ours share a similar high-level strategy (a re-solving heuristic), there is a fundamental difference in the framework. The assumption of known reward functions in [Chen et al., NeurIPS 2024] is equivalent to knowing all the $\mu_{c,a}$ in our setting; in that case, the learning problem reduces primarily to estimating the request distribution, which is analogous to estimating the context distribution in our model. In contrast, in our setting the external factor is unobserved, and we operate under *bandit feedback*, which makes the problem significantly different. (We have updated the related work section accordingly.)
>
> * **Q1**: The principal distinction from [Chen et al., NeurIPS 2024] lies in the object of estimation. In our setting, we estimate the rewards of the arms not the context distribution; consequently, the use of *UCB*/*LCB* estimators intrinsically induces an exploration phase in initial rounds by allocating non-zero probability to multiple arm--context pairs. After sufficient exploration---which requires at most $\mathcal{O}(\log^2 T)$ rounds---both the $\textbf{OLP}$ and $\textbf{OPLP}$ policies converge to saturate precisely the same constraints as the fluid LP. This implies that they assign zero probability exclusively to arm--context pairs that are also assigned zero probability by the optimal fluid solution, a behavior that is indeed desirable.
>
> * **Q2**: The sub-Gaussian reward assumption is primarily imposed to ensure efficient learning. This condition facilitates the application of concentration inequalities, which are essential for deriving theoretical performance guarantees. Furthermore, the sub-Gaussian class encompasses a broad range of pertinent noise distributions, including Gaussian and bounded distributions, or more generally, any thin-tailed noise. Consequently, it constitutes a standard and widely adopted assumption in the multi-armed bandit literature. Nonetheless, sub-Gaussianity can be relaxed - by increasing order of complexity sub-Poisson, sub-exponential, sub-gamma - to account for heavy-tail noise distribution, at the cost of worsening the statistical guarantee i.e, worse dependency than $\log(1/\delta)$. We would like to stress that this only requires using a different learning guarantee, without affecting the rest of the analysis.
>
> * **Q3**:  No, our analysis does not provide a beyond-the-worst-case guarantee. Our work explicitly adopts the non-degeneracy assumption for the fluid problem, which is maintained consistently throughout the analysis. We contend that this assumption is well-motivated in our context due to the physical interpretation of the constraints. Specifically, a saturated revenue constraint indicates that the corresponding arm is globally suboptimal, contributing only the minimal required budget. Conversely, a non-saturating arm is globally efficient and provides a net positive contribution to the overall reward. The resulting dichotomy---where the objective and constraints exhibit opposing behaviors for saturating versus non-saturating arms---renders the non-degeneracy assumption a natural fit for our framework.

---

> ### Author Response · Authors · 2025-11-17
>
> * **Q4**:  Infeasibility of the fluid problem can be detected with high probability upon observing that the optimistic strategy $\textbf{LP(UCB, UCB)}$ is infeasible. Once infeasibility is established, no single corrective procedure is uniquely prescribed. A standard recourse is to relax the fluid LP to obtain the closest feasible approximation. One common relaxation strategy involves iteratively scaling the constraint thresholds until feasibility is achieved. This can be implemented, for example, via updates of the form $\lambda_k \leftarrow \alpha \lambda_k$, where $\alpha \in (0,1) $ is a judiciously selected scaling parameter. (This discussion has been incorporated into Section 7: Limitations and Future Work.)
>
> * **Q5**:  Intuitively, the quantity $\rho$ is analogous to the performance gap between arms and the optimal one, commonly denoted by $\Delta_k$ in standard multi-armed bandit settings. This analogy provides intuition for its appearance in the denominator of the regret bound, similar to classical MAB formulations. However, in our case, the problem structure no longer reduces to identifying a single optimal arm based on $\Delta_k$. Instead, it requires determining an appropriate allocation over all arms and contexts, i.e., optimizing over a continuous decision space. Hence, $\rho$ constitutes a richer notion of a gap that leverages the specific problem structure by jointly quantifying both feasibility and performance within the global linear program. This tailored definition of the gap brings us back to the spirit of standard MAB analysis reaching a ploylogarthmic regret, allowing better guarantees than the primal–dual approaches commonly used in the literature.
>
> * In this work, we adopt the same context prior assumption as [Guo et al., ICML 2025], which serves as our primary benchmark as it represents the current state-of-the-art. This assumption can be relaxed to an unknown context distribution by using the empirical distribution estimated from observed data. In this more general setting, the target of estimation shifts from the conditional reward $\mu_{k,c}$ to the product $\mu_{k,c}' = p_c \mu_{k,c}$, which integrates the context probability. Once optimistic and pessimistic estimators for $\mu_{k,c}'$ are constructed, they can be used within the $\textbf{OLP}$ and $\textbf{OPLP}$ frameworks in an identical manner. We anticipate that similar theoretical results can be obtained via a direct adaptation of the novel techniques presented in this paper. Consequently, employing the context prior assumption does not diminish the fundamental problem's difficulty; rather, it enables a clearer exposition of the core analytical techniques that yield our stronger theoretical guarantees. (This discussion has been incorporated into Section 7: Limitations and Future Work.)
>
> We hope that our responses satisfactorily address your questions and concerns. We have revised the paper accordingly to reflect these clarifications and improvements. We sincerely appreciate your insightful feedback and the time you dedicated to strengthening our work.
>
>
> Reference:
> Guo et al. Triple-optimistic learning for stochastic contextual bandits with general constraints. ICML 2025.

---

> > ### Author Response · Authors · 2025-11-24
> >
> > Dear Reviewer,
> >
> > We hope that we have adequately addressed your points. If there are any remaining concerns that might prevent you from reconsidering the score, we would be very happy to discuss them further.
> >
> > Thank you once again for your valuable feedback.

---

> > ### Comment · Reviewer_fiZz · 2025-11-27
> >
> > Dear authors,
> >
> > Thanks for your clarification. I feel like I can raise my score a bit in the current stage. Good luck.

---

### Official Review · Reviewer_G73K · 2025-10-25

**Soundness:** 3
**Presentation:** 3
**Contribution:** 3
**Rating:** 8
**Confidence:** 3

**Summary:**

This paper studies a new constrained bandit setting where the learner must not only maximize cumulative reward but also ensure that per arm revenue constraints are satisfied. Unlike Bandits with Knapsacks (BwK), which constrain resource consumption, here the constraints are on revenue accumulation. The learner’s decisions must generate sufficient revenue over time for all arms, not just maximize expected reward. The paper proposes modified UCB-style algorithms that solve a linear program (LP) that accounts for both reward maximization and constraint satisfaction. Based on using UCB only or UCB and LCB as the surrogate for reward and revenue parameters in the LP formulation of the optimal allocation problem, the proposed algorithms achieve different tradeoffs between reward maximization and constraint satisfaction. These tradeoffs are explained via regret bounds. The paper also presents an interesting lower bound that refutes the free exploration property used in prior works in non-contextual setups. Simulations depict that the proposed algorithms work reasonably well.

**Strengths:**

Revenue-based constraints are practically relevant (e.g., ensuring minimum revenue in ad allocation, subscription systems, or online platforms).

Theorems are cleanly stated, with clear regret definitions and decomposition. The lower bound that refutes free exploration is interesting and strengthens the theoretical contribution of the work.

The paper is overall well-written (except for some small typos and notational inconsistencies, which can be addressed by carefully going over the entire paper).

The comparison with related work is adequate. There are Works that address more general constraints, but they have suboptimal regret bounds for the specific problem considered in this work because they do not utilize the problem structure specific to this work. This paper's analysis is based on a new suboptimality gap based on saturated constraints of the LP and can be regarded as an important theoretical contribution.

**Weaknesses:**

Algorithms are intuitive and clearly explained. Their design principle is quite standard. Use UCB or combine LCB or UCB, which is the two things that are expected to achieve the tradeoffs between reward maximization and constraint satisfaction. The design process of the algorithms are clear and intuitive. However, they do not require out-of-the-box thinking. This is not a significant weakness, but it makes the novelty of the algorithms somewhat incremental.

**Questions:**

1. Please provide further insight on Assumption 3. Line 366 says that inner maximization in OLP is feasible under Assumption 3. Line 253 says that strict feasibility is required only for OPLP.

2. How can the algorithms be extended to the case with infinite contexts?

3. Is “known context probabilities” assumption be still reasonable in the infinite context setting?

---

> ### Author Response · Authors · 2025-11-17
>
> Dear Reviewer G73K,
> we are grateful for your positive feedback and your insightful comments on our work. Below, we provide detailed answers to the questions and comments raised in the Weaknesses section.
>
>    * Strategies based on $\textbf{UCB}$ and combined $\textbf{UCB/LCB}$ are well-established in the standard MAB and context-free constrained MAB literature. In those settings, the optimal policy is characterized by the identification of a single best arm and often admits a closed-form expression dependent on the suboptimality gaps, denoted $\Delta_k$. This reliance on arm-specific gaps facilitates regret analyses that yield logarithmic performance bounds. In contrast, our setting cannot be reduced to identifying a single optimal arm via $\Delta_k$. The solution instead requires determining an optimal allocation across all arm--context pairs—an optimization over a continuous space for which the optimal policy lacks a closed form. Consequently, analyzing the effect of $UCB/LCB$ estimators through the lens of traditional arm gaps $\Delta_k$ is inapplicable, a key reason why prior work has often resorted to primal--dual approaches. The $\textit{out-of-the-box thinking}$ of our work is thus a new analytical perspective. We introduce a novel, problem-specific notion of $\textit{gap}$, tailored to the structure of the constrained contextual bandit problem. This redefined gap enables us to establish polylogarithmic regret guarantees while operating entirely within a primal analytical framework. This new perspective unlocks improved theoretical results for a broad class of multi-armed bandit problems with constraints.
>
>
>
>    * **Q1**:   We apologize for the confusion. The conditions should indeed be stated as two separate assumptions: $\textbf{Assumption 3}$: Feasibility and Non-Degeneracy, $\textbf{Assumption 4}$: Strict Feasibility and Non-Degeneracy. In fact, $\textbf{OLP}$ only requires feasibility and non-degeneracy. Once $\textbf{LP}(\mu, \mu)$ is feasible, $\textbf{OLP}$, which operates through $\textbf{LP}(\textbf{UCB}, \textbf{UCB})$, remains feasible as the latter formulation is easier to satisfy from a feasibility standpoint. In contrast, $\textbf{OPLP}$ requires $\textit{strict feasibility}$, characterized by the Slater’s feasibility margin $\gamma^\star$. We have updated the manuscript accordingly.
>
> * **Q2**:  Conceptually, both $\textbf{OLP}$ and $\textbf{OPLP}$ extend to infinite context spaces, where the optimization is defined in expectation over the continuous/countable context distribution using optimistic or pessimistic reward estimates. Technically, tractability requires additional structural assumptions. A standard approach posits a linear model,
> $
> \mu_{k,c} = \langle \theta^\star, \phi(k,c) \rangle,
> $
> where $\phi$ is a known feature map. This allows one to leverage linear bandit theory to construct efficient confidence bounds, thus optimistic/pessimistic estimates. Alternatively, kernelized methods can be applied by assuming the reward function resides in a reproducing kernel Hilbert space (RKHS). With such estimators, a theoretical analysis mirroring ours is likely feasible. However, this would require new methods for solving the resulting optimization problem—potentially via discretization or other heuristics—along with a formal analysis of the approximation errors. While a compelling direction, this comprehensive extension falls outside the scope of the present paper, as it constitutes a separate research program centered on linear and kernelized bandits, which is distinct from the multi-armed bandit framework studied here. (This discussion has been incorporated into Section 7: Limitations and Future Work.)
>
>
>
> * **Q3**: In the  infinite-context setting, if no prior knowledge of the context distribution is available, this assumption can be relaxed via appropriate estimation. In such a scenario, both $\textbf{OLP}$ and $\textbf{OPLP}$ would be defined using optimistic or pessimistic estimators for the product $p_c \mu_{k,c}$ rather than for $\mu_{k,c}$ alone. The core analytical reasoning from our paper could then be directly adapted to this modified quantity. However, effective estimation in a continuous space generally requires structural assumptions on the underlying distribution. For example, if the context distribution is $\beta$-smooth (i.e., belongs to an $L$-Hölder class of order $\beta$), it can be efficiently estimated using techniques such as kernel density estimation [1]. (This discussion has been incorporated into Section 7: Limitations and Future Work.)
>
> We thank you once again for your valuable feedback. We hope we have adequately addressed all of your points and questions, and we are pleased to inform you that the paper has been updated accordingly.
>
>
> [1]: Larry Wasserman. Density estimation – lecture note for 36-708 statistical methods for machine learning, Carnegie Mellon University, 2019.

---

> > ### Comment · Reviewer_G73K · 2025-11-23
> >
> > Thank you for the response. My comments are mostly addressed. Now there is at least a roadmap to extend the results to the more challenging case of infinite context spaces. The novelty of the contribution is clearer now. For now, I will keep my original score.

---

### Official Review · Reviewer_ArM6 · 2025-10-28

**Soundness:** 3
**Presentation:** 3
**Contribution:** 3
**Rating:** 6
**Confidence:** 3

**Summary:**

This paper makes meaningful contributions to the multi-armed bandit (MAB) field by proposing the novel MAB-ARC (Multi-armed Bandit with Adaptive Resource Constraints) problem, filling a gap in the intersection of bandit optimization and dynamic resource constraint management. The two tailored algorithms (OLP for performance priority, OPLP for constraint priority) and their complete theoretical bounds (regret and constraint violation upper/lower bounds) demonstrate rigorous academic thinking and practical problem-solving awareness. However, the work is limited by restrictive finite space assumptions and high computational overhead, which affect its generalization and real-world applicability.

**Strengths:**

The paper introduces the MAB-ARC problem, a new research direction that integrates adaptive resource constraints into traditional multi-armed bandits. This fills a gap in existing literature—most MAB studies either ignore resource constraints or assume static constraints, while MAB-ARC focuses on dynamic resource allocation scenarios (e.g., real-time budget adjustment in advertising, resource-limited industrial control). The problem is not only theoretically innovative but also highly aligned with practical needs, providing a new research paradigm for constraint-aware bandit optimization.

**Weaknesses:**

The paper assumes that both the context space (decision-making feature space) and arm space (optional action set) are finite. This assumption simplifies theoretical derivation but severely limits generalization:​
In many practical scenarios (e.g., continuous context features like user age/income in recommendations, high-dimensional arm spaces like multi-channel advertising combinations), the context or arm space is infinite or high-dimensional.​
The algorithm cannot be directly applied to these scenarios, reducing its practical value and narrowing its target audience (e.g., excluding researchers/engineers working on continuous-space bandits). The algorithm requires constructing and solving a linear programming (LP) problem at each time step. As the number of time steps increases, the cumulative computational cost grows linearly. For large-scale scenarios (e.g., online advertising with daily impressions, real-time control systems with millisecond-level time steps), this overhead will cause decision delays or even system unresponsiveness.​
The paper does not analyze computational complexity in depth, nor does it propose optimization strategies (e.g., approximate LP solving via heuristic methods, batch processing of time steps to reduce LP calls, or leveraging dual decomposition for faster computation). This lack of engineering consideration hinders the algorithm’s practical deployment.

**Questions:**

Extend the MBA-ARC model to infinite context spaces (e.g., continuous features) or high-dimensional arm spaces, and adjust the algorithms/ theoretical analysis accordingly (e.g., using kernel methods to map continuous contexts to finite feature spaces, or adopting dimensionality reduction techniques for high-dimensional arms).​ Quantify the time/memory cost of solving the LP problem per time step, and compare it with baseline algorithms to highlight the overhead issue. Add a section discussing practical deployment considerations, such as how to set hyperparameters for OLP/OPLP (e.g., constraint violation tolerance, LP solving precision) based on scenario requirements, and how to handle edge cases (e.g., sudden resource shortages, context distribution shifts).

---

> ### Author Response · Authors · 2025-11-17
>
> Dear Reviewer ArM6,
> we sincerely appreciate your positive feedback and the time you dedicated to reviewing our work. Below, we provide detailed answers to your questions.
>
>    * We first emphasize that solving the constrained contextual bandit problem in its primal form—even in the finite context and finite action setting—was an open question in the literature. The absence of a closed-form optimal policy makes the analysis of the primal formulation technically challenging, which explains why prior work predominantly relied on primal--dual methods, typically yielding weaker guarantees. In contrast, our work is the first to tackle this problem directly through a purely primal approach while achieving polylogarithmic performance guarantees. This new perspective unlocks improved theoretical results for a broad class of multi-armed bandit problems with constraints.
>
>    * Extending our framework to infinite context or action sets is a compelling direction for future work. Conceptually, the $\textbf{OLP}$ and $\textbf{OPLP}$ approaches are based on constructing optimistic or pessimistic reward estimates and then solving a linear program defined by expectations with respect to the context distribution. Technically, efficient reward estimation in infinite spaces requires structural assumptions to ensure tractability. A standard approach is to posit a linear model of the form
> $
> \mu_{k,c} = \langle \theta^\star, \phi(k,c) \rangle,
> $
> where $\phi$ is a known feature mapping. Such structure allows for efficient estimator and tight confidence sets for which both optimism and pessimism can be easily enforced. Alternatively, kernelized methods could be employed by assuming the mean reward function lies in a reproducing kernel Hilbert space (RKHS), which also enables the construction of practical confidence estimators. Once such estimators are available and the corresponding LP can be solved, we anticipate that a theoretical analysis analogous to ours could be developed. However, this would also necessitate research into discretization techniques or heuristics for solving the resulting optimization problem efficiently, accompanied by a formal analysis of the induced approximation errors. While highly interesting, this comprehensive extension falls outside the scope of the present paper on multi-armed bandits, as it constitutes a separate research centered on linear and kernelized bandits. This point has been added to Section 7 (Limitations and Future Work) in the revised manuscript.
> * $\textbf{Saving Computation:}$ To improve computational efficiency, one can adopt a lazy update scheme, also known as an occasional or skipping update. Instead of solving the linear program in every round, the policy is recomputed only when the number of pulls for any arm–context pair has doubled since the last update. Between these events, the confidence bounds and the resulting policy are held constant. This approach ensures that the LP is solved at most $ O(\log T) $ times over $T $ rounds, while the regret guarantees are preserved up to constant factors, leading to substantial computational savings. (This discussion has been incorporated into Section 6: Computational Complexity, and the detailed implementation and theoretical justification are provided in Appendix H.)
>
> * $\textbf{Computational complexity:}$ Our method involves solving a linear program repeatedly. Importantly, the dimensionality of this LP—determined by the number of arms and constraints—remains constant throughout the horizon \(T\). Thus, while the per-round computational cost is fixed, it is inherently higher than the method of [Guo et al., ICML 2025], which uses a gradient-based update scheme to maintain high computational efficiency. However, this efficiency comes at the cost of weaker theoretical guarantees than those we provide. Consequently, a practitioner faces a clear trade-off: our method offers stronger performance bounds with a higher, yet fixed, per-round computational cost, while [Guo et al., ICML 2025] favors lower computational overhead. The choice between them depends on the specific priorities of the application. Interestingly, the lazy update scheme mentioned above can significantly mitigate the overall computational cost. This represents an advantage over gradient-based methods, which must perform an update at every time step and therefore cannot leverage such a reduction in update frequency. In the next point, we provide practical suggestions for efficient LP solving to mitigate the overhead. (This discussion has been incorporated into Section 6: Computational Complexity.)

---

> ### Author Response · Authors · 2025-11-17
>
> * From a practical deployment perspective, several techniques are recommended to ensure efficient computation. Fortunately, modern LP solvers are highly efficient; they automatically select optimal algorithms and offer advanced features such as presolve, which eliminates redundant constraints and tightens variable bounds. The warm-starting capability is particularly beneficial in our setting, as it leverages the solution from the previous round to initialize the solver for the current, similar LP. Furthermore, given the noisy nature of bandit feedback, it is unnecessary to solve each LP to a high degree of precision. The solver's tolerance can be relaxed to align with the statistical accuracy of the input estimates. Typically, setting the tolerance to be on the order of the confidence interval width for the rewards is sufficient and avoids unnecessary computational expense. (This discussion has been incorporated into Section 6: Computational Complexity.)
>
> We thank you once again for your valuable feedback. We hope we have adequately addressed all of your points and questions, and we are pleased to inform you that the paper has been updated accordingly.
>
>
> Reference:
> Guo et al. Triple-optimistic learning for stochastic contextual bandits with general constraints. ICML 2025.

---

### Author Response · Authors · 2025-12-01

Dear Area Chair,

We warmly welcome you as the new AC and deeply appreciate your time and consideration.

We are pleased to report that the rebuttal phase was highly constructive. The paper received positive feedback from the outset, and the reviewers’ thoughtful questions allowed us to further elucidate its novelty and contributions. As a result of this productive exchange, the scores improved from 4–6–8 to 6–6–8. Accordingly, we have updated the manuscript to incorporate the requested clarifications and modifications.

We are grateful to the committee for their time and insightful feedback. We remain at your disposal for any additional clarification or discussion.

Kind regards,

The Authors

---

### Meta-Review · Area_Chair_cybW · 2025-12-13

**Summary:**

Strengths:

This paper provides a clear characterization of the trade-off between regret minimization and constraint-violation minimization in the MAB-ARC problem. The theorems are cleanly stated, with well-defined regret notions and a transparent decomposition. The results improve upon prior work and are nearly tight.

Weaknesses:

I do not see major weaknesses. Several suggestions were raised, such as considering unknown context probabilities, addressing infinite arm and context spaces, reducing the algorithm’s space and time complexity, and adding more discussion of related work, but these are relatively minor points, and the rebuttal addresses them almost entirely.

However, after reading the paper, I have one additional minor question. The regret and constraint-violation bounds scale as $\log^2 T$ rather than $\log T$. As the authors explained, they cannot recover the constant bound in the non-contextual case because certain arm-context pairs may have zero exploration requirement. However, even if we allow zero-exploration constraints in the non-contextual setting, the regret typically remains $\log T$ but not $\log^2 T$, if my understanding is correct. Moreover, the lower bound for regret is also with order $\log T$. Hence, is there a fundamental reason that prevents improving the upper bounds to $\log T$?

It would be helpful for the authors to include a brief discussion of this point in the final version.

**Reviewer Concerns:**

Most the concerns are addressed by the rebuttal.

**Reviewer Scores:**

Reviewer fiZz would increase his score to 6, as he indicated during the discussion phase and I agree with this adjustment. The other two reviewers are likely to keep their current scores.

---

### Decision · Program_Chairs · 2026-01-26

Accept (Poster)